# Beyond Pretrained Features:
# Noisy Image Modeling Provides Adversarial Defense

**Zunzhi You**[1], **Daochang Liu**[1], **Bohyung Han**[2], **Chang Xu**[1]
[1] School of Computer Science, University of Sydney
[2] ECE & IPAI, Seoul National University
zyou5834@uni.sydney.edu.au, daochang.liu@sydney.edu.au,
bhhan@snu.ac.kr, c.xu@sydney.edu.au

## Abstract

Recent advancements in masked image modeling (MIM) have made it a prevailing framework for self-supervised visual representation learning. The MIM pretrained models, like most deep neural network methods, remain vulnerable to adversarial attacks, limiting their practical application, and this issue has received little research attention. In this paper, we investigate how this powerful self-supervised learning paradigm can provide adversarial robustness to downstream classifiers. During the exploration, we find that noisy image modeling (NIM), a simple variant of MIM that adopts denoising as the pre-text task, reconstructs noisy images surprisingly well despite severe corruption. Motivated by this observation, we propose an adversarial defense method, referred to as De$^3$, by exploiting the pretrained decoder for denoising. Through De$^3$, NIM is able to enhance adversarial robustness beyond providing pretrained features. Furthermore, we incorporate a simple modification, sampling the noise scale hyperparameter from random distributions, and enable the defense to achieve a better and tunable trade-off between accuracy and robustness. Experimental results demonstrate that, in terms of adversarial robustness, NIM is superior to MIM thanks to its effective denoising capability. Moreover, the defense provided by NIM achieves performance on par with adversarial training while offering the extra tunability advantage. Source code and models are available at https://github.com/youzunzhi/NIM-AdvDef.

## 1 Introduction

The idea of fine-tuning pretrained deep neural networks is essential for the success of machine learning to date. A deep neural network that has been well-pretrained on a large-scale dataset can serve as a good initialization for various downstream tasks and improve their performance. Within this paradigm, self-supervised learning plays a vital role as this approach allows the networks to take advantage of vast amounts of unlabeled data. With the emergence of powerful, scalable Transformer-based vision models [14, 21], the masked image modeling (MIM) has rapidly developed recently and become the new dominant paradigm for visual feature pretraining [38, 17, 8, 26], following the success of masked language modeling [12, 4] in NLP. Conceptually, the idea behind MIM is simple. During the pretraining phase, unlabeled training images are randomly masked patch-wise and then fed into an encoder-decoder architecture, where the decoder attempts to recover the original images from the features embedded by the encoder.

Although pretraining methods like MIM leads to improved standard performance on downstream visual tasks, the adversarial robustness of the fine-tuned model is not enhanced. The vulnerability to imperceptible yet maliciously crafted perturbations will significantly limit the real-world deployment of MIM, especially for safety-critical applications. Although the existing state-of-the-art approach,

37th Conference on Neural Information Processing Systems (NeurIPS 2023).

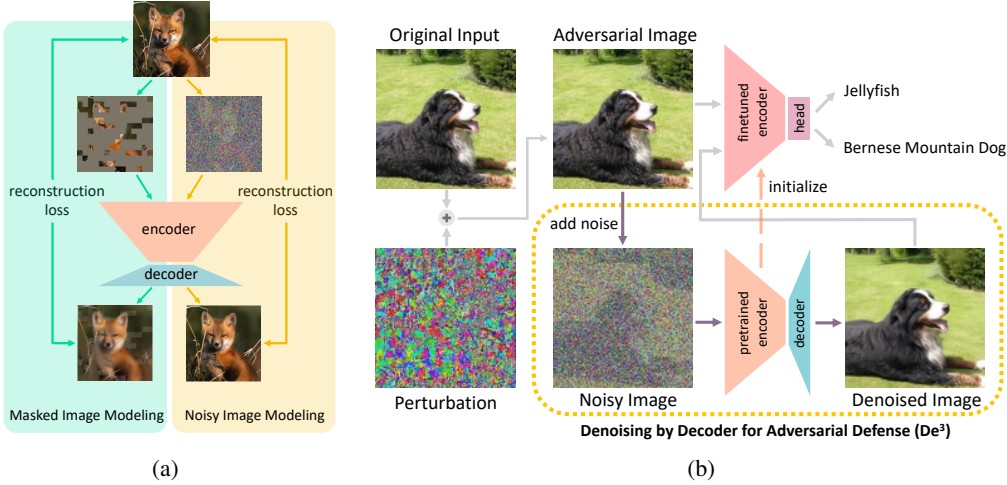

Figure 1: (a) Comparison between the NIM framework and the MIM framework. The two frameworks can be almost exactly the same except for the degradation step. (b) An illustration of the proposed De$^3$ adversarial defense. De$^3$ first adds a certain level of noise to the attacked image, then uses the model pretrained by the NIM framework to denoise the image.

adversarial training [22, 23], can train adversarially robust models, it is known to be computationally expensive [22] and can result in a significant drop in accuracy on clean data [34], which undermines the usefulness of the MIM pretrained features. Moreover, compared to standard training, the sample complexity of achieving adversarially robust generalization is significantly higher, increasing the importance of label efficiency in practice. Motivated by these concerns, we ask the question: *How can the state-of-the-art generative pretraining paradigm provide adversarial robustness in addition to standard accuracy*?

As we explore this question, we observe two notable characteristics about the paradigm: (1) Other degradations than masking patches, such as zooming-in, blurring, or masking in the frequency domain, can be effective for the pretext task of self-supervised learning [33, 37]. (2) An attempt to understand MIM revealed that it implicitly learns occlusion-invariant features [19]. Motivated by the observations, we explore Noisy Image Modeling (NIM), a variant of MIM that uses denoising as the pretext task and learns to encode noise-invariant features (Figure 1 (a)).

During the exploration, we find that NIM's denoised images are of surprisingly high quality, even from intensely noisy images (Figure 3 (e) and (g)). However, in the common practice of MIM, only the encoder is retained for further finetuning, while the decoder that performs image reconstruction is discarded. Therefore, the strong denoising capability of NIM will be wasted, which we believe can be utilized for removing adversarial perturbations. To address this, we propose an adversarial defense method named De$^3$: **De**noising by **De**coder for Adversarial **De**fense, which first adds some Gaussian noise to the adversarial samples and then tries to reconstruct the original images, as shown in Figure 1 (b). With De$^3$, NIM provides not only pretrained features but also an adversarial defense to its fine-tuned models. We further propose to randomize the hyperparameter that controls the degradation level in pretraining, so the defense provided by NIM can achieve a tunable trade-off between standard accuracy and adversarial robustness, which further enhances its applicability in practice.

To summarize, this work has the following contributions:

- We propose a novel method called De$^3$ to utilize the strong denoising ability of NIM models so they can provide defense against adversarial attacks beyond pretrained features.

- Rather than setting the hyperparameter that controls the degradation level globally, as done in existing MIM approaches, we propose to sample it from random distributions, so the defense achieves a flexible accuracy-robustness trade-off that can be tuned at test time.

- Extensive experiments show that our NIM pretraining learns as good visual features as MIM, while being advantageous for it effectively enhances the robustness of fine-tuned models against strong adversarial attacks.

## 2   Related Work

### 2.1   Self-supervised Pretraining in Vision

Learning visual representations from unlabeled images has been a research direction in machine learning and computer vision that has drawn increasing attention in recent years. While masked language modeling (MLM) has achieved great success in pretraining large-scale language models [12, 4, 20], the mainstream framework for pretraining vision models was contrastive learning (CL) [6, 18, 7] until recently. With the significant emergence of Transformers adapted for vision tasks [14, 5, 21], the counterpart of MLM in vision, MIM, becomes the new prevailing self-supervised pretraining paradigm [26, 13, 17, 38]. Through the process of masking and predicting image patches, the encoder learns to extract meaningful visual features.

Beyond using mask prediction as the pretext task, there have been studies examining the outcomes of employing alternative degradation methods. Tian *et al.* [33] investigate five methods, including zoom-in, zoom-out, distortion, blurring, and de-colorizing, and discover that all of these methods outperform supervised pretraining. Additionally, Xie *et al.* [37] explore masking in the frequency domain and demonstrate that similar low-level tasks such as super-resolution and denoising yield competitive performance. These studies inspire us to explore the framework of noisy image modeling. More recently, Fang *et al.* [15] also modify the masking operation in the generative pretraining framework, where the input images are degraded by an additional generator with a small trainable BEiT [2]. However, instead of solely using the generative pretraining framework for pretrained features, we further propose to leverage it for enhancing adversarial robustness.

### 2.2   Adversarial Robustness

Since about a decade ago when researchers found that deep neural networks are fragile to imperceptible adversarial perturbations [32, 3], enhancing their robustness to such attacks has been an active research area, as stronger threat models have also been continuously developed [16, 25, 22, 10]. One of the most successful methods, adversarial training (AT), incorporates adversarial examples during training to enhance the model's adversarial robustness [16, 22, 39]. While effective, AT methods present many challenges that limit their applicability. For instance, due to the necessity of searching for effective attacks for each image during training, the computational cost can be too expensive. Moreover, AT achieves high robustness at the cost of compromising standard accuracy [34, 27]. Although Wang *et al.* [36] mitigate this issue by a model-conditional training approach that enables an in-situ calibration of the trade-off between robustness and accuracy, the training cost is further increased, and the method is limited to CNNs.

Another line of research aims to defend against adversarial attacks in the input space, as opposed to the model space like AT. These methods often use an additional generator to restore clean images from attacked ones, such as GAN [29] or Diffusion Models [24]. As their training does not depend on the threat model like AT, these methods can often defend against unseen threats. However, their performance is usually not comparable with AT methods [10]. Our De$^3$ method is distinct from these methods because our defense is provided by pretraining, eliminating the need for an additional generative model. Therefore, our cost of obtaining the defense also brings pretrained features and boosts fine-tuning. More importantly, the goal of this work is mainly to show the advantage of NIM over MIM in terms of adversarial robustness, not to propose a novel adversarial defense method.

## 3   Noisy Image Modeling

### 3.1   Preliminary: Masked Image Modeling

We first briefly revisit the MIM pretraining framework. For every image $\mathbf{x} \in \mathbb{R}^{H \times W \times C}$, where $H$, $W$, and $C$ denote the height, width, and channel dimensions respectively, sampled from the training dataset $\mathbf{\Omega}$, MIM first randomly samples a binary mask $\mathbf{M}$ under the control of the mask ratio

hyperparameter $\gamma$ so that $\sum_i^N m_i/N = \gamma$, where each element $m_i \in {0,1}$ of the binary mask $\mathbf{M}$ corresponds to a pixel in the image, and $N$ is the total number of pixels. Then, the masked image is given by

$$\hat{\mathbf{x}}_{MIM} = \mathbf{x} \odot (\mathbf{1} - \mathbf{M}), \qquad (1)$$

where $\odot$ denotes the element-wise product. After that, MIM uses its encoder $f_\theta$ and decoder $g_\phi$ parameterized by $\theta$ and $\phi$, respectively, to predict the reconstructed image. In practice, the encoder $f_\theta$ of MIM is usually instantiated by Vision Transformers (ViTs) [14] or its variants [21], where patch-wise masks are employed. Finally, MIM adopts some distance metric $\mathcal{D}(\cdot, \cdot)$ such as $\ell_1$ loss to measure the reconstruction quality of the masked patches. Overall, the objective of MIM is formulated as follows:

$$\min_{\theta,\phi} \mathbb{E}_{\mathbf{x} \in \boldsymbol{\Omega}} \mathcal{D}(g_\phi(f_\theta(\hat{\mathbf{x}}_{MIM})) \odot \mathbf{M}, \mathbf{x} \odot \mathbf{M}). \qquad (2)$$

### 3.2 Noisy Image Modeling: Learning Noise-invariant Feature

Figure 1 (a) presents the similarity between the process of MIM and NIM pretraining. As a variant of MIM, NIM shares many components including the network architecture design, the reconstruction target, and the training techniques. The only difference is that NIM degrades the original image by adding Gaussian noise as

$$\hat{\mathbf{x}}_{NIM} = \mathbf{x} + \sigma\epsilon, \quad \epsilon \sim \mathcal{N}(0, \mathbf{I}), \qquad (3)$$

where $\sigma$ is the parameter that controls the degradation level, i.e., the noise scale. The objective function of NIM is given by

$$\min_{\theta,\phi} \mathbb{E}_{\mathbf{x} \sim \boldsymbol{\Omega}} \mathcal{D}(g_\phi(f_\theta(\hat{\mathbf{x}}_{NIM})), \mathbf{x}). \qquad (4)$$

As MIM learns to predict masked regions, the encoder-decoder pretrained under NIM is able to denoise corrupted input images. Note that the term "noise" we use has a different meaning from that in the classical Denoising autoencoders (DAE) [35]. In our context, we refer to the Gaussian noise that is *added* to the image, while DAE corrupts the input by replacing randomly selected pixels with 0, which is actually equivalent to masking..

Following the work trying to understand MIM from an occlusion-invariant feature learning perspective [19], we present how NIM learns noise-invariant features. Assume there exists a decoder network $g'_{\phi'}$ parameterized by $\phi'$ that can almost restore the feature embedded by $f_\theta$ to the its input image:

$$g'_{\phi'}(f_\theta(\mathbf{x})) \approx \mathbf{x}. \qquad (5)$$

Then, the objective function of NIM in Eq. (4) can be rewritten as

$$\min_{\theta,\phi} \mathbb{E}_{\mathbf{x} \sim \boldsymbol{\Omega}} \mathcal{D}(g_\phi(f_\theta(\hat{\mathbf{x}}_{NIM})), g'_{\phi'}(f_\theta(\mathbf{x}))). \qquad (6)$$

By defining a new similarity measurement

$$\mathcal{D}'_{\phi,\phi'}(z_1, z_2) \triangleq \mathcal{D}(g_\phi(z_1), g'_{\phi'}(z_2)), \qquad (7)$$

Eq. (6) is further simplified as

$$\min_{\theta,\phi} \mathbb{E}_{\mathbf{x} \sim \boldsymbol{\Omega}} \mathcal{D}'_{\phi,\phi'}(f_\theta(\hat{\mathbf{x}}_{NIM}), f_\theta(\mathbf{x})). \qquad (8)$$

Hence, the objective function of NIM can be viewed as minimizing the distance of feature embedding between the original image $\mathbf{x}$ and the noisy image $\hat{\mathbf{x}}_{NIM}$, implying that the feature embedding learned by $f_\theta$ should be noise-invariant.

## 4 De$^3$: Denoising by Decoder for Adversarial Defense

### 4.1 Formulation of De$^3$

Although MIM is effective in representation learning, it is still challenging to learn robust pretrained features. Fine-tuned models are easily compromised by small adversarial perturbations, limiting their use in safety-critical tasks. With the awareness of this problem, we explore the potential of NIM and find that the reconstruction ability of the NIM models is surprisingly good. For example, Figure 3 (e) and (g) illustrate the reconstructed images from the noisy images. We observe that most

of the semantics are preserved in the reconstruction even for the images that are too noisy to be recognized by humans. However, the common practice of the MIM pretraining only uses the encoder for the initialization of downstream models while the decoder that has the outstanding reconstruction capability is simply discarded. Meanwhile, we believe that the remarkable denoising capability of the NIM decoder can be employed to enhance the adversarial robustness of downstream models and propose a simple defense technique referred to as De[3]: **De**noising by **De**coder for Adversarial **De**fense.

As shown in Figure 1 (b), De[3] works during the test time of downstream tasks by first adding Gaussian noise to an input image and then denoising the image using the pretrained decoder as follows:

$$y = c_\psi(g_\phi(f_\theta((\mathbf{x}' + \sigma_{De^3}\epsilon)))), \quad (9)$$

where $c_\psi$ is the fine-tuned downstream network parameterized by $\psi$ given the pretrained $\theta$, $\mathbf{x}'$ is the input image that may be under adversarial attacks, and $\sigma_{De^3}$ is the noise scale used for the defense. The motivation behind this method is straightforward: since the adversarial perturbation is bounded to a small $\epsilon$, it should be easily flooded by the Gaussian noise with a much larger magnitude and then can be removed along with the random Gaussian noise in the decoding process. Also, the noise is randomly generated during testing, which is usually not fully accessible to the threat models even in white-box attack settings, making it more difficult to find a way for an effective attack.

While the reconstruction of NIM models yields realistic results with minimal error, there remains a discernible gap between the reconstructed images and the original input images. Therefore, we propose to fine-tune the downstream model with denoised images, where training examples used for fine-tuning $c_\psi$ also undergo an adding-noise-then-denoising procedure, simulating the conditions encountered during test time. The objective function of fine-tuning is formulated as follows:

$$\min_\psi \mathbb{E}_{\mathbf{x} \in \mathbf{\Omega}} \mathcal{L}(c_\psi(g_\phi(f_\theta(\mathbf{x} + \sigma_{ft}\epsilon))), \bar{y}), \quad (10)$$

where $\mathcal{L}$ is the loss function, $\bar{y}$ is the ground-truth label of $\mathbf{x}$, and $\sigma_{ft}$ is the noise scale using in the fine-tuning. Note that $\theta$ and $\phi$ are fixed during the fine-tuning process.

### 4.2 Achieving Tunable Trade-off by Sampling Random $\sigma$ in Pretraining

Intuitively, the proposed De[3] method should have a tunable trade-off between clean accuracy and adversarial robustness by adjusting $\sigma_{De^3}$. Increasing $\sigma_{De^3}$ will make adversarial perturbations more likely to be flooded and subsequently removed, resulting in stronger adversarial robustness. Meanwhile, as the degradation becomes more severe, the reconstruction quality diminishes, leading to lower clean accuracy However, we observe that this assumption does not necessarily hold in practice. Figure 2 (a) demonstrates that reconstruction quality deteriorates when the noise levels imposed on input images during training and testing are different. We further show how the reconstruction loss[1] varies with noise levels in Figure 2 (b) and observe the same phenomenon in the pretrained models with globally set $\sigma = 25, 75$, and $150$. These observations suggest that if the model is exposed only to a single noise level during training, it struggles to generalize its denoising capability to images with different noise levels.

To address this issue, we propose to sample the noise level parameter $\sigma$ from a random distribution instead of fixing it globally as existing MIM methods do. Figure 2 illustrates that models pretrained with randomly selected $\sigma$'s can effectively reconstruct input images with good quality, even for corrupted images across various noise scales, as long as the scale is not excessively large. Therefore, we can identify a good trade-off between accuracy and robustness in the NIM models.

## 5 Experimental Results

In this section, we empirically show the effectiveness of NIM as a self-supervised pretraining method that can bring adversarial robustness via the De[3] method. We describe the settings in Section 5.1 and compare NIM with MIM in Section 5.2. Then, in Section 5.3, the proposed adversarial defense De[3] using NIM is compared with the adversarial training approach. Finally, Section 5.4 presents the ablation study that discusses how $\sigma$ in pretraining makes differences to the model performance. More experimental results and details can be found in the supplementary.

---

[1]Lower loss indicates better reconstruction.

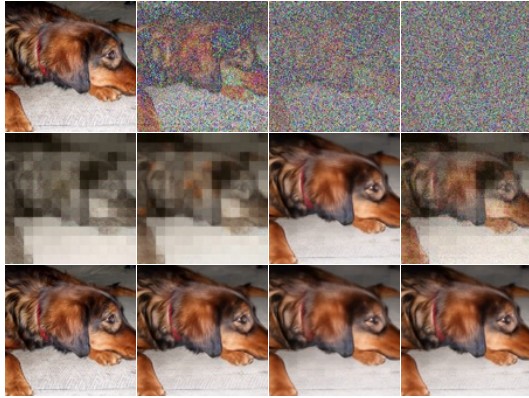
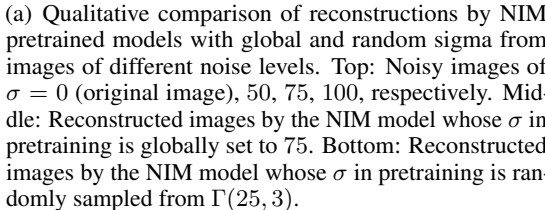

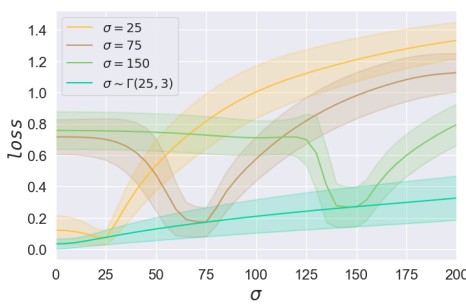

(a) Qualitative comparison of reconstructions by NIM pretrained models with global and random sigma from images of different noise levels. Top: Noisy images of $\sigma = 0$ (original image), 50, 75, 100, respectively. Middle: Reconstructed images by the NIM model whose $\sigma$ in pretraining is globally set to 75. Bottom: Reconstructed images by the NIM model whose $\sigma$ in pretraining is randomly sampled from $\Gamma(25, 3)$.

(b) Relationship between the reconstruction loss and the noise level $\sigma$ of models pretrained with $\sigma$ set to 25, 75, 150 globally and sampled from $\Gamma(25, 3)$. The standard deviation is also shown.

Figure 2: Qualitative and quantitative comparisons show that NIM pretraining with globally set $\sigma$ only reconstructs well from the noisy images of noise level close to the $\sigma$ in pretraining, while NIM pretraining with random $\sigma$ reconstructs well for any $\sigma$ in a reasonable span.

## 5.1 Experimental Setups

**Datasets and Backbones**  We conduct all the experiments on ImageNet-1K [11] dataset. We use the training set (1.28 million images) in pretraining and finetuning and use the validation set (50,000 images) for evaluation. We adopt ViT-Base (ViT-B/16) [14] as the backbone in our main experiments. Results for additional backbones can be found in the supplementary.

**Threat Models**  To evaluate the adversarial robustness, we adopt untargeted $\ell_\infty$-bounded attacks with radius $\epsilon = 4/255$, which is the most commonly used setting for adversarial robustness studies on ImageNet. We consider three popular white-box attacks: single-step attack FGSM [16], multi-step attack PGD [22], and the state-of-the-art attack AutoAttack (AA) [10]. For PGD, we set the number of steps $n = 10$ and step size $\alpha = 2/255$. For AA, we use its 'rand' version since our defense is a randomized defense. To save the computational cost, we apply the AA attack on a 5000-image subset of the ImageNet1K validation set selected by RobustBench [9].

**Training Implementations**  We adopt two representative MIM methods, MAE [17] and Sim-MIM [38], as the baseline MIM methods. By modifying only the degradation part, we train our NIM-MAE and NIM-SimMIM: the pretext task is changed from mask prediction (with a globally set mask ratio) to denoising (with randomly sampled $\sigma$), while all the other hyperparameters and training techniques follow the original paper. Our default models are pretrained with $\sigma \sim \Gamma(25, 3)$ and finetuned on denoised images of $\sigma \sim U(0, 30)$. All models are pretrained for 800 epochs and then finetuned for 100 epochs.

## 5.2 Comparing NIM with MIM

In Table 1, we compare our NIM models to the MIM baselines. First, we show that without any adversarial defense, the NIM-pretrained classifiers are slightly less accurate but more robust than the MIM-pretrained classifiers. Therefore, we believe that even in settings where a defense model is unavailable, NIM is a simple but effective self-supervised visual learning framework and is worth more investigation in the future.

Table 1: Comparisons of ImageNet-1K clean and robustness accuracies (%) between MIM and NIM-MIM, with or without the proposed De$^3$ defense. $^\dagger$De$^3$ refers to the defense method that adapts De$^3$ to MIM methods (i.e., *de-masking* by decoder for adversarial defense). Numbers following De$^3$ indicate the degradation level used in defense, where $\gamma$ is the mask ratio and $\sigma$ is the noise scale. NIM-MAE models that use De$^3$ for defense are finetuned on denoised images from the pretrained model.

| Model | MAE [17] | | NIM-MAE | | | SimMIM [38] | | NIM-SimMIM | | |
|---|---|---|---|---|---|---|---|---|---|---|
| De$^3$ | None | $\gamma$=0.75 | None | $\sigma$=70 | $\sigma$=140 | None | $\gamma$=0.6 | None | $\sigma$=70 | $\sigma$=140 |
| Clean | **83.05** | 44.96 | 82.58 | 78.68 | 70.69 | **83.62** | 43.05 | 82.76 | 76.23 | 68.76 |
| FGSM | 28.29 | 36.38 | 31.66 | **53.66** | 53.12 | 30.38 | 34.29 | 31.90 | **51.84** | 51.67 |
| PGD-10 | 0.25 | 11.58 | 0.31 | 34.61 | **39.82** | 1.13 | 7.39 | 2.02 | 33.41 | **37.45** |
| AA | 0.00 | 2.58 | 0.00 | 23.24 | **33.70** | 0.00 | 1.82 | 0.00 | 21.70 | **32.46** |

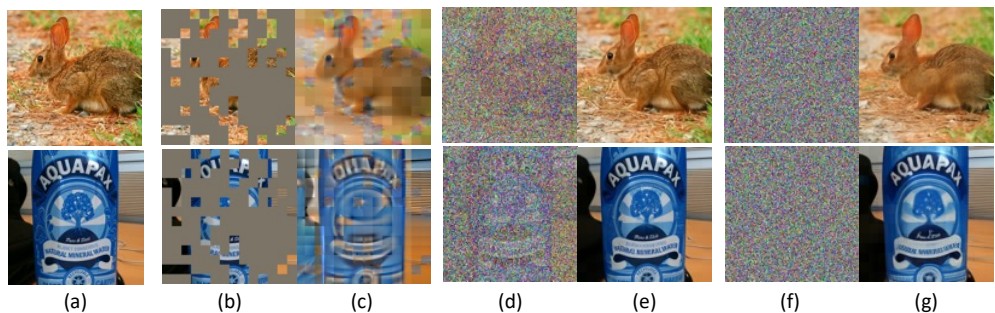

| (a) | (b) | (c) | (d) | (e) | (f) | (g) |

Figure 3: Qualitative results of reconstruction by MAE and NIM-MAE pretrained models. (a) Original images. (b) 75% masked images. (c) Reconstructed images of (b) by the MAE pretrained model. (d) Images added $\sigma$ =70 Gaussian noise. (e) Reconstructed images of (d) by our NIM-MAE pretrained model. (f) Images added $\sigma$ =140 Gaussian noise. (g) Reconstructed images of (f) by our NIM-MAE pretrained model. The images shown are randomly selected from the ImageNet-1K validation set.

With the De$^3$ defense provided by the NIM pretrained models, the downstream finetuned classifiers obtain strong robustness for all white-box attacks. Numerically, when the Gaussian noise's $\sigma$ is 70 (or 140), the magnitude of the added Gaussian noise is about $70/4 = 17.5$ (or 35) times the adversarial noise. Therefore, the adversarial noise can be flooded and removed along the Gaussian noise by De$^3$. As a result, the NIM models with De$^3$ outperform their MIM counterparts without defense with up to 25.37% improvement in terms of robustness against FGSM attacks, while only 4.37% clean accuracy is decreased. The De$^3$ defense is also effective against strong attacks like PGD-10 and AA that almost 100% successfully make the vanilla models fail.

In contrast, MIM does not benefit from the defense. Columns 2 and 7 are the results of using the MIM pretrained models for adversarial defense. Here, the De$^3$ defense is adapted for MIM models: it is performed by first randomly masking some patches and then reconstructing the masked images via the pretrained encoder and decoder, where the mask ratio remains the same as in pretraining (75% for MAE and 60% for SimMIM). It is shown that MIM pretrained models are unable to offer effective adversarial defenses as the clean accuracy drops drastically while robustness improves marginally. Figure 3 shows the qualitative results of using MAE and NIM-MAE to reconstruct masked and noisy images, respectively. Noticeably, the reconstruction of MAE is much worse than NIM-MAE, even when the $\sigma$ is 140 and the images are unidentifiable for humans. We further quantitatively evaluate the reconstruction quality by computing PSNR (peak signal-to-noise ratio, higher the better) between the original and reconstructed images over the whole ImageNet-1K validation set. As a result, the PSNR of MAE is 20.88 when the mask ratio is 75%, while for NIM-MAE, it is 27.43 when $\sigma$ =70 and 24.9 when $\sigma$ =140.

Table 2: Comparisons of ImageNet-1K clean and robustness accuracies (%) between adversarial trained ViT-base models and NIM-MAE model with the $De^3$ defense.

| Pretrain | Defense | Clean | FGSM | PGD-10 | AA |
|---|---|---|---|---|---|
| scratch | Adv. Training | 56.97 | 39.89 | 24.25 | 19.50 |
| MAE | Adv. Training | 69.67 | 51.80 | 39.65 | 34.56 |
| NIM-MAE | Adv. Training | 69.28 | 52.77 | **39.97** | **34.84** |
| NIM-MAE | $De^3$ ($\sigma = 140$) | **70.69** | **53.12** | 39.82 | 33.70 |

## 5.3 Comparing NIM+$De^3$ with Adversarial Training

To further help understand the effectiveness of our NIM models with the $De^3$ defense, we compare it with the adversarial training method in this section. Adversarial training for ViTs is not a trivial task and techniques and hyperparameters for CNNs may not be beneficial or applicable for ViTs [1, 30]. Here, we adopt the adversarial training recipe for ViTs provided by a recent work [23]. We do the PGD-5 adversarial training for 20 epochs so the training time is close to ours and adjust the learning rate schedule accordingly. Table 2 shows the results of doing adversarial training from scratch and the MAE and NIM-MAE pretrained models. It is shown that both MAE and NIM-MAE are helpful for the adversarial training of ViTs. Compared to the adversarial models, our NIM-MAE model with $\sigma$=140 $De^3$ defense shows comparable robustness with slightly higher clean accuracy.

Adversarially trained models often compromise the clean accuracy too much. In our experiment, the performance for natural images downgrades by 13.38% for the MAE model. Although this issue can be mitigated by setting a larger weight for the standard loss term in the objective function [39], the adjustment of the trade-off comes at the cost of starting over another training process. In contrast, our $De^3$ defense enables the testing time tunable trade-off, i.e., we do not need to train another network to trade robustness for accuracy (or vice versa). By increasing the scale of noise added in the defense at test time, more adversarial perturbations can be flooded and removed by the denoising pretrained model, at the expense of poorer reconstruction quality and a greater loss of semantic information (Figure 3 (e) vs. (g)), resulting increased robust accuracies and decreased clean accuracy (Figure 4). For example, our model's robustness against the PGD-10 attack can be improved from almost 0 to 31.25% and only compromise 2% of clean accuracy (79.37%). Therefore, in practice, our defense can adjust its performance on clean and adversarial images according to the requirement of the context. For example, in autonomous driving, the need for adversarial robustness may increase when the vehicle comes into an adverse environment, while for safer places, the need for high clean accuracy is prioritized.

While we prepare this paper, some more recent studies on adversarial training for ViTs have emerged [28, 31], and the adversarial robustness of ViTs on ImageNet-1K was improved significantly with more delicate implementation and longer training time. Although our NIM with $De^3$ approach does not achieve the same level of adversarial robustness as these existing works, we want to highlight that the aim of this work is not to compete with the state-of-the-art methods. Instead, we aim to demonstrate to the research community that NIM can serve as a promising and advantageous self-learning paradigm for enhancing adversarial robustness.

## 5.4 $\sigma$ in Pretraining

In this section, we present our ablation studies to evaluate how the random distributions of the Gaussian noise scale parameter $\sigma$ in pretraining influence the fine-tuned models' performance. We adopt MAE as the default implementation of MIM and randomly selected 5,000 images when evaluating robust accuracy for reducing experimental overhead.

First, we present the clean and robust accuracy of models without defense. The left subfigure of Figure 5 shows that NIM pretrained models can achieve competitive accuracy on clean images, especially when the degradation level hyperparameter is set to or concentrated at a fair value (i.e., $\sigma = 75$, $\sigma \sim \Gamma(25, 3)$, $\sigma \sim U(50, 100)$). On the other hand, most of the NIM models lead to higher adversarial robustness than MAE, if the variance of $\sigma$ is not too large (e.g., $\sigma \sim U(0, 250)$, $\sigma \sim \Gamma(3, 25)$. Such phenomenon is consistent with the understanding that NIM learns noise-invariant features in Section 3.2.

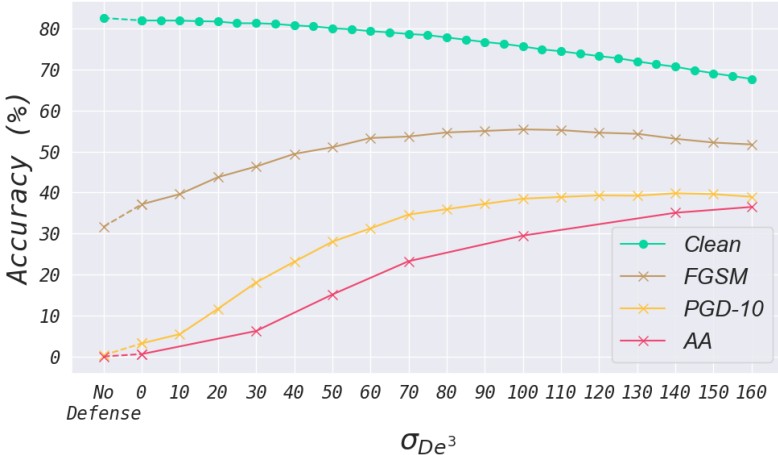

Figure 4: Clean and robust accuracies of our NIM-MAE with De$^3$ of different $\sigma$ . The clean accuracy decreases and robust accuracies increase as the $\sigma$ increases, enabling a tunable trade-off in testing time.

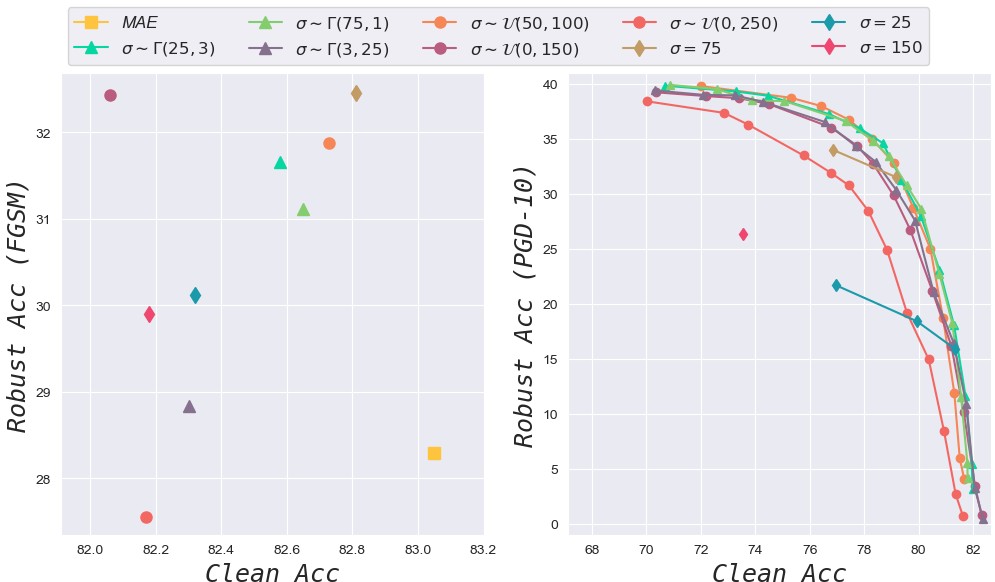

Figure 5: Performance of MAE and NIM-MAE with different $\sigma$ in pretraining. Left: Accuracy on clean images and adversarial images attacked by FGSM of MAE and NIM-MAE fine-tuned models without De$^3$ defense. Right: The Pareto frontiers of NIM-MAE models with De$^3$ defense between clean accuracy and robust accuracy against PGD-10 attacks. The Pareto frontiers are measured by using different $\sigma$ in defense (increased by 10 from 0 to 160).

In the right subfigure of Figure 5, we show the empirical Pareto frontier of De$^3$ using different NIM-MAE pretrained models, i.e., the points on the figure are the models' best achievable trade-offs between clean and robust accuracy by using different $\sigma$ in defense. We first observe that the Pareto frontiers of models using globally set $\sigma$ collapse to very few points because using $\sigma$ other than the one in pretraining leads to lowering both clean and robust accuracy. Meanwhile, the results suggest that $\sigma$ sampled from the Gamma distribution generally achieves better accuracy-robustness trade-off than the ones from the Uniform distribution. Among all models, $\sigma \sim \Gamma(25, 3)$ achieves the best performance by being concentrated to a moderate value and having a suitable variance.

# 6  Conclusions

In this paper, we investigate noisy image modeling (NIM), a simple variant of masked image modeling where the pretext task is changed from mask prediction to denoising. We discover that NIM exhibits a remarkable ability to reconstruct images even from images with intense noise and are motivated to utilize this denoising capability to remove adversarial perturbations. We propose a simple method that enables the NIM models to provide not only pretrained features but also an adversarial defense. To further achieve a tunable trade-off during test time, we sample the noise level hyperparameter from random distributions rather than setting it globally as MIM. We demonstrate by extensive experimental results that in terms of adversarial robustness, NIM is advantageous over MIM thanks to its strong denoising ability. We also show that the adversarial defense provided by NIM achieves comparable performance to a recent adversarial training method and offers greater flexibility and applicability due to its tunable accuracy-robustness trade-off. We hope that our work will motivate the community to explore NIM and other variants of MIM so the full potential of generative visual pretraining can be realized.

# Acknowledgment

This work was supported in part by the Australian Research Council under Projects DP240101848 and FT230100549, and also in part by the NRF and IITP grants funded by the Korea government (MSIT) (No. 2022R1A2C3012210, 2021-0-01343).

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
