## A  Broader Impacts

As ViTs and MIM become the most popular deep learning approaches in computer vision, their susceptibility to adversarial attacks raises concerns regarding their applications in safety-critical tasks like autonomous driving. To tackle this issue, our research delves into the potential of leveraging MIM to enhance the adversarial robustness of downstream models. By improving the robustness of ViTs, our method contributes to the development of more robust and trustworthy modern computer vision systems. However, it is crucial to note that our results indicate that our model still falls short of achieving complete robustness against adversarial attacks, which means it remains vulnerable to manipulation by malicious actors. Additionally, the additional denoising step introduced in our approach may cause negative environmental impacts, as it may result in increased energy consumption and contribute to carbon emissions. But, as our method uses pretraining models as the adversarial defense, it can be more energy-saving in training time than other methods, such as adversarial training.

## B  Limitations

One major drawback of our method is that the defense process, involving the addition of noise followed by denoising, introduces a computational overhead and increases inference time. This aspect can be particularly significant when real-time performance is a crucial requirement for the task at hand.

It is important to highlight that our paper's focus is specifically on the adversarial robustness of ViTs. We have not evaluated other variants of Transformers, such as the Swin Transformer (2), within the scope of this work. Additionally, while our defense mechanism proves effective against noise-based adversarial attacks, it may not exhibit robustness against other common corruption forms, such as snow, blur, or pixelation.

## C  Scalability of NIM

We show the evaluation results of using different backbone model sizes and training scheme lengths in Figure 1. As its MIM variant, MAE (1), our NIM-MAE models also achieve higher performance by scaling up the model size and increasing the number of training epochs.

## D  Robustness under More Attacks

Next, we present our model's performance under different white-box PGD attacks. Figure 2 shows the robust accuracies under attacks of different perturbation budgets and optimization steps, respectively. It is shown that our method can provide an effective defense against severe adversarial attacks. Even for the threat with $\epsilon = 8/255$, our model can achieve 22.95% top-1 robust accuracy while retaining 70.69% clean accuracy.

## E  Analysis for NIM with De³

We propose two hypotheses for explaining the reason behind our method's effectiveness: (1) Given an adversarial image, the NIM pretrained model is able to remove the adversarial noise along with the Gaussian noise added; (2) The Gaussian noise added is randomly sampled and unknown to the attacker, making it difficult for the threat model to find the effective attack. To test the hypotheses, we conduct an experiment where the attacker has access to the Gaussian noise used in the defense process. Note that this setting is typically unrealistic in the real world as the noise is sampled during inference time. Figure 3 (a) shows the comparison between the results of noise being known and unknown. When the attacker can access the noise, our model's robust accuracy does not improve much as $\sigma$ increases. Therefore its highest robust accuracy is much lower but not as low as it is without the defense. The results indicate that both proposed hypotheses are true.

Figure 3 (b) and Figure 4 show the quantitative and qualitative results of our model's reconstruction from adversarial images of different $\epsilon$. It is observed that the reconstruction gets worse as $\epsilon$ increases, indicating that besides the adversarial perturbation, the low-quality reconstruction caused by adversarial attacks could also be a reason for robust accuracy being lower than clean accuracy.

# F   More Experimental Results on MIM

In this section, we show more results of using MIM pretrained models as an adversarial defense. We first do an ablation study on the pretraining objective. All models are pretrained for 800 epochs with ViT-Base as the backbone architecture.

From the recovered images of MIM models (Figure 4 to Figure 5 of (4), Figure 2 to Figure 4 of (1), Figure 3 of our main paper), we notice that reconstruction in the unmasked area is worse than the masked because the prediction target is limited to the *masked* patches:

$$\min_{\theta,\phi} \mathbb{E}_{\mathbf{x}\in\mathbf{\Omega}} \mathcal{D}(g_\phi(f_\theta(\hat{\mathbf{x}}_{MIM}) \odot \mathbf{M}, \mathbf{x} \odot \mathbf{M}), \tag{1}$$

Therefore, we experiment with models whose reconstruction target is for *all* patches:

$$\min_{\theta,\phi} \mathbb{E}_{\mathbf{x}\in\mathbf{\Omega}} \mathcal{D}(g_\phi(f_\theta(\hat{\mathbf{x}}_{MIM}), \mathbf{x}), \tag{2}$$

Besides, we also experiment with a randomized mask ratio $\gamma \sim \mathcal{U}(0,1)$ as random $\sigma$ improves the performance for our approach. We only use the SimMIM framework for this ablation since masked patches are not fed to the encoder in MAE, and as a result, the mask ratio has to be consistent for each sample within a single batch.

Figure 6 shows the PSNR of reconstructed images by different pretrained models from masked images of different $\gamma$. It is shown that models targeted to reconstruct all patches have much higher PSNR than their counterparts that only recover the masked patches. However, results in Figure 5 suggest that pretrained MIM models recovering all patches still cannot provide effective defense despite that the clean accuracy is improved. Note that here we evaluate the robust accuracy by adopting an easier PGD-10 attack (3) with $\epsilon = 2/255$. Although the MAE model that recovers all patches achieves a better accuracy-robustness trade-off than the default model, it still underperforms our NIM model. Moreover, unlike NIM models, the performance of using the model pretrained with random $\gamma$ as defense is inferior to the one that uses globally set $\gamma$.

We also conduct experiments that finetune the model with reconstructed images. Here we use the MAE model targeted to reconstruct all patches as the default pretrained model. The results are summarized in Table 1. In contrast to NIM models, it is shown that finetuning on reconstructed images does not boost the performance of MIM models as an adversarial defense.

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

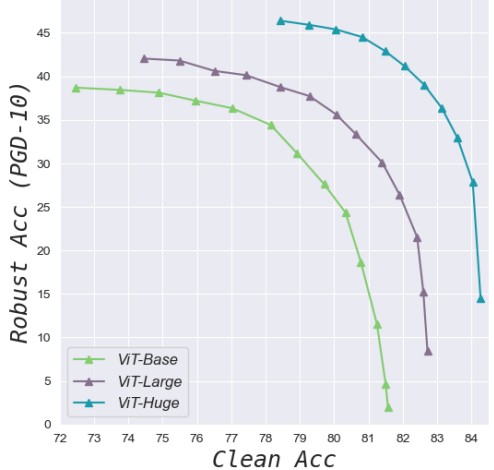
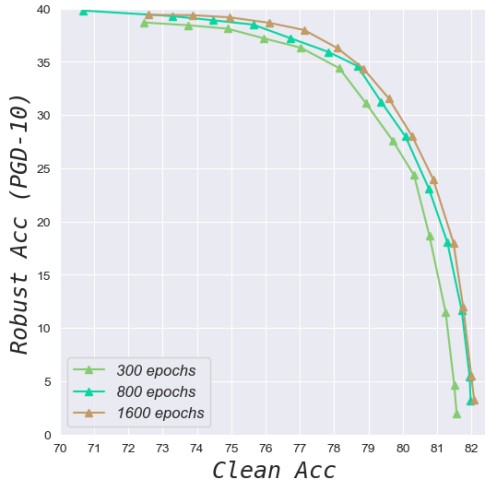

(a) The Pareto frontiers of NIM-MAE models with different backbone sizes. Each model is pretrained for 300 epochs with $\sigma \sim \Gamma(25, 3)$. Following (1), ViT-Base is fine-tuned for 100 epochs, while ViT-Large and ViT-Huge are fine-tuned for 50 epochs.

(b) The Pareto frontiers of NIM-MAE models with different training scheme lengths. Each model is from a full training schedule and uses ViT-Base as the backbone.

Figure 1: The Pareto frontiers of NIM-MAE models show that the NIM models can scale up model sizes and training with a longer time improves the performance.

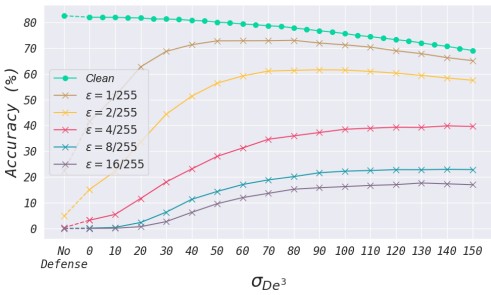
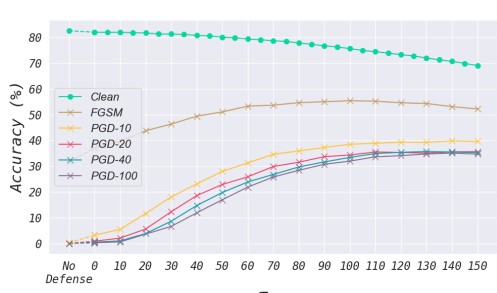

(a) Robust accuracies under PGD-10 attacks with different $\epsilon$.

(b) Robust accuracies under FGSM and PGD attacks with different steps ($\epsilon = 4/255$).

Figure 2: The robustness of our NIM-MAE with $\text{De}^3$ with different $\sigma$ under more white-box untargeted attacks.

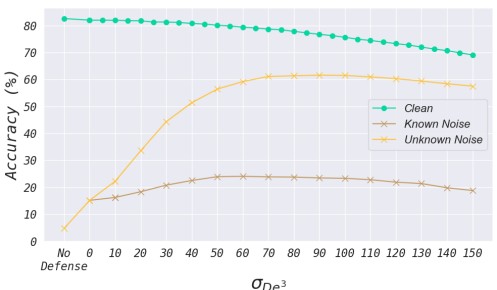 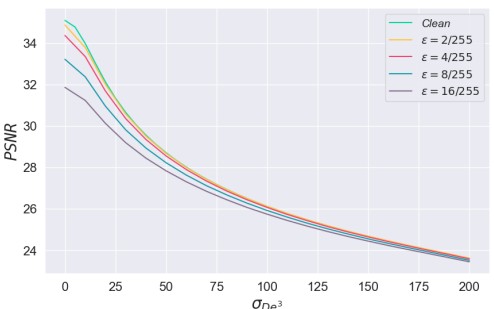

(a) Robust accuracies under PGD-10 attacks ($\epsilon = 2/255$) that has access to the exact sampled noise used in the defense, compared to the one under the same attack that has full access to the gradients of the models and the defending strategy (defending $\sigma$) but no access to the noise.

(b) PSNR of reconstructed images from clean and adversarial samples of different $\epsilon$. The PSNR is computed against the clean images.

Figure 3: Quantitative results for the analysis of our NIM with De$^3$ defense.

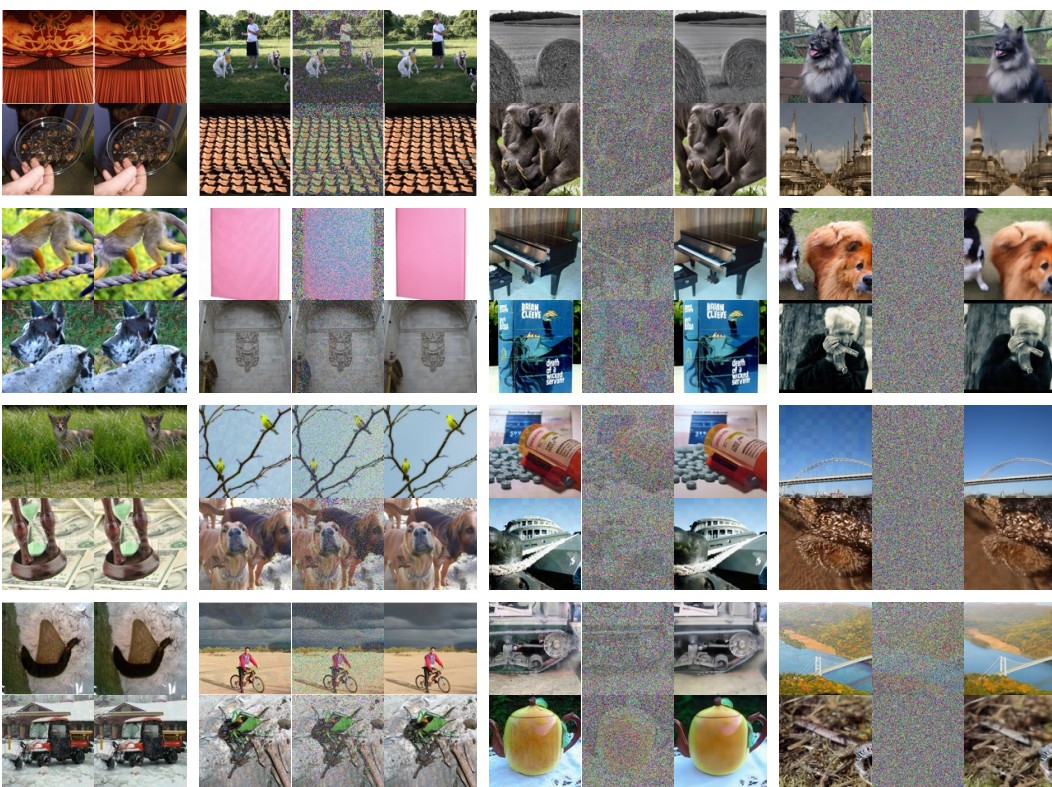

Figure 4: Reconstruction results of our NIM-MAE model from adversarial images attacked by PGD-10. From left to right, the noise level $\sigma$ of the degraded images is 0, 30, 75, and 150, respectively. For each triplet, the adversarial image (left), the degraded image (middle), and the reconstruction (right) are shown. The degraded images are omitted for $\sigma = 0$ since they are equivalent to the adversarial images. From top to bottom, the perturbation budget $\epsilon$ is 2/255, 4/255, 8/255, 16/255, respectively. The images shown are randomly selected from ImageNet-1K validation set.

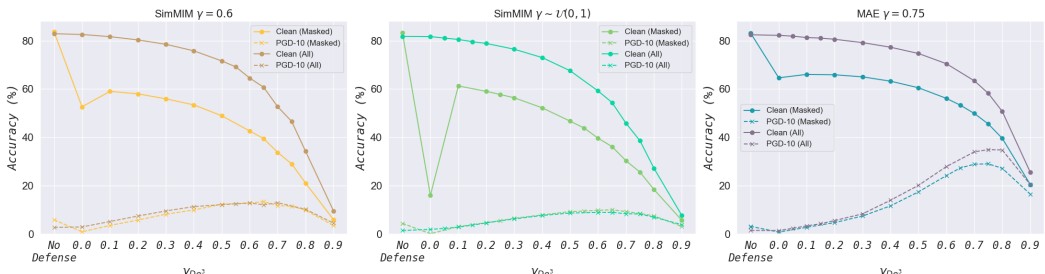

Figure 5: Clean and robust accuracies of different MIM models using De$^3$ with different masking ratios $\gamma$ as an adversarial defense. The robust accuracy is evaluated against the PGD-10 attack with $\epsilon = 2/255$.

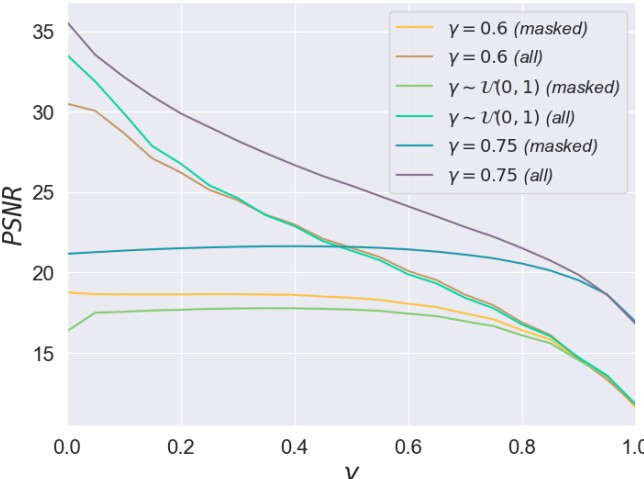

Figure 6: The Relationship between the PSNR of reconstructed images and the mask ratio $\gamma$ of different MIM pretrained models with different pretraining $\gamma$.

Table 1: Ablation experiments on $\gamma$ used in finetuning. Models are pretrained with the MAE (1) framework that uses all patches as the reconstruction target. The clean and robust accuracy of the defense $\gamma$ that gets the highest robust accuracy is shown. The robust accuracy is evaluated against the PGD-10 attack with $\epsilon = 2/255$.

| Finetune $\gamma$ | Clean | WB |
|---|---|---|
| Original | 58.20 | 34.90 |
| $\sim \mathcal{U}(0, 15)$ | 46.40 | 29.57 |
| $\sim \mathcal{U}(0, 30)$ | 48.91 | 31.43 |