# OpenReview forum: "Beyond Pretrained Features: Noisy Image Modeling Provides Adversarial Defense"
_NeurIPS.cc/2023/Conference — NeurIPS 2023 poster_

### Official Review · Reviewer_qarJ · 2023-06-29

**Soundness:** 2 fair
**Presentation:** 2 fair
**Contribution:** 2 fair
**Rating:** 3
**Confidence:** 5

**Summary:**

This paper proposed a denoising pre-training framework NIM similar to MIM to improve adversarial defense ability. In particular, the input image is a noisy version of the original. The pre-training goal is to do denoising.

**Strengths:**

This paper is easy to follow.

**Weaknesses:**

 * a) Novelty is limited. The idea proposed in this work is very similar to CIM [1]. Compared to MIM, this work changes the masking operation to add noise.

* b) Performance is incremental with limited applicability. The proposed De3 as a pure self-supervised pre-training approach does not outperform MIM baselines (MAE and SimMIM). For downstream adversarial robustness tests, De3 shows improvement over MAE and SImMIM, which is not surprising since the pre-training goal is to denoise.

* c) Datasets and backbones used in this work to validate performance are scarce. It is suggested that more datasets,downstream tasks, and backbones are evaluated.

**References**

[1]  Fang, Yuxin, et al. "Corrupted image modeling for self-supervised visual pre-training." ICLR 2023.

**Questions:**

See weakness.

**Limitations:**

See weakness.

---

> ### Author Rebuttal · Authors · 2023-08-05
>
> Thank you for your valuable feedback. We'd like to address your concerns:
>
> > Novelty is limited. The idea proposed in this work is very similar to CIM [1].
>
> There are many major differences between our work and [1], including
> 1) Purpose: Our goal is to explore how the generative pretraining paradigm can provide adversarial robustness beyond pretrained visual features, while the focus of [1] is to learn better pretrained visual representations.
> 2) Method: We propose a novel method that utilizes the strong denoising ability of NIM models to remove the adversarial perturbation, while [1] proposes to adopt an additional generator with a small trainable BEiT to degrade the input image.
> 3) Conclusions: We show that NIM is able to achieve a strong and tunable accuracy-robustness trade-off that MIM models are unable to compete, indicating the superiority of NIM over MIM in terms of adversarial robustness. In contrast, [1]’s authors demonstrate that both ViTs and CNNs can learn rich visual representations using a unified framework.
>
> Overall, the core novelty of our work lies in the proposed methods that effectively leverage the pretraining NIM models for providing adversarial defense to downstream models, which is recognized by other reviewers.
>
> > Performance is incremental with limited applicability.
>
> We respectfully disagree that our method’s performance is incremental. As explicitly stated in our title, this work aims to go beyond pretrained features and see if pretrained models can provide adversarial robustness to downstream models. We show that our proposed method achieves a significant improvement in robustness against PGD attack by 34.36%, with a marginal clean accuracy drop of 4.37%. Even without the $De^3$ defense, the NIM model’s robust accuracy against FGSM attack is higher by 3.37% than MAE, while the clean accuracy of NIM-MAE is slightly lower by 0.47%.
>
> > De3 shows improvement over MAE and SImMIM, which is not surprising since the pre-training goal is to denoise.
>
> First, we would like to argue that setting the target improvement as the training goal is a natural and reasonable idea in machine learning research, which would not diminish the work’s value. For example, adversarial training [2] incorporates adversarial examples in training. While it might seem "unsurprising" that adversarial trained models are robust to adversarial examples, the importance of this work has been universally acknowledged. In the context of our work, we believe what matters is that NIM’s improvement over MIM is under fair settings: from pretraining and fine-tuning to evaluation, we ensure everything is the same and the only variable is the degradation method.
>
> On the other hand, we believe our results are interesting for that our training goal is not exactly the target: we train the model to denoise *Gaussian noise*, instead of specific *adversarial noise*. Therefore, at inference time, we propose to add Gaussian noise whose magnitude is much larger than the imperceptible adversarial noise, so the latter can be flooded and removed along. From this perspective, the idea is interesting and instructive, as appreciated by fellow reviewers.
>
> > Datasets and backbones used in this work to validate performance are scarce.
>
> Following the suggestion, we conducted another experiment on CIFAR-10, based on the implementation of [3]. We set the attack radius $\epsilon = 8/255$, and other settings remain the same as in Table 1 of our main paper. The results are as follows:
>
> | Model | MAE | MAE | NIM-MAE | NIM-MAE | NIM-MAE |
> | --- | --- | --- | --- | --- | --- |
> | $De^3$ | None | $\gamma$=0.75 | None | $\sigma$=40 | $\sigma$=70 |
> | Clean | **89.88** | 78.36 | 88.31 | 81.09 | 76.30 |
> | FGSM | 17.51 | **65.13** | 21.66 | 43.26 | 48.86 |
> | PGD-10 | 0.01 | 22.50 | 2.77 | 32.24| **40.26** |
> | AA | 0.00 | 6.220 | 0.00 | 31.49 | **41.20** |
>
> Interestingly, we observe that unlike on ImageNet, MAE provides a stronger defense against FGSM on CIFAR-10. It is likely because, 1) for low-resolution images, MAE can generate good reconstructions from masked images; 2) when using MAE in $De^3$, 75% of adversarial perturbations will be masked, and if the attack is weak like FGSM, the rest of the perturbation would be too weak to break the model. However, for stronger attacks, NIM still provides better defense than MAE.
>
> Regarding more downstream tasks, we would like to remind the reviewers that implementing adversarial attacks for other tasks than classification is not a trivial work [4, 5] and is beyond the scope of this paper. To the best of our knowledge, researchers normally would only conduct experiments on image classification to show the effectiveness of an adversarial defense [6, 7, 8].
>
> Regarding the backbones, we would like to draw attention to Figure 1(a) in the supplementary material, where we have demonstrated that our method is scalable and effective with larger backbones.
>
> ---
> [1] Fang, et al. Corrupted image modeling for self-supervised visual pre-training. ICLR 2023.
>
> [2] Madry, et al. Towards Deep Learning Models Resistant to Adversarial Attacks. ICLR 2018.
>
> [3] https://github.com/IcarusWizard/MAE.
>
> [4] Croce, et al. Robust Semantic Segmentation: Strong Adversarial Attacks and Fast Training of Robust Models. 2023.
>
> [5] Agnihotri, et al. CosPGD: a unified white-box adversarial attack for pixel-wise prediction tasks. 2023.
>
> [6] Rebuffi, et al. Revisiting adapters with adversarial training. ICLR 2023.
>
> [7] Mo, et al. When adversarial training meets vision transformers: Recipes from training to architecture. NeurIPS 2022.
>
> [8] Yoon, et al. Adversarial Purification with Score-based Generative Models. ICML 2021.

---

> > ### Comment · Reviewer_qarJ · 2023-08-15
> >
> > I thank the authors for their response. However, I still have the following concerns:
> >
> > * CIM is a very representative work in the MIM domain. Since it is very similar to the method proposed by the authors, at least it should be cited. Moreover, different research purpose does not distinguish the two works. For instance, applying the idea of MAE to audio pre-training is an application but not an innovation.
> >
> > * The pre-training objective of CIM is to denoise the corrupted input image, which is exactly the same idea proposed by the authors in this work. However, although the architectural designs and corruption approaches might be different, the underlying idea is the same.
> >
> > * Adding Gaussian noise to the input image and asking the model to denoise during pre-training improves the model's robustness towards noise in general.
> >
> > * More results on datasets with various scales and types are required to draw solid conclusions. For instance, IN-21k, iNaturalist, COCO etc.
> >
> > With these concerns, I will stand by my rating.

---

> > > ### Author Response · Authors · 2023-08-16
> > > **Further discussion with Reviewer qarJ (Part 1/2)**
> > >
> > > > CIM is a very representative work in the MIM domain. Since it is very similar to the method proposed by the authors, at least it should be cited.
> > > >
> > >
> > > While we acknowledge that CIM is a representative MIM variant, we would like to emphasize again that it has little similarity with our proposed method. In CIM, the authors proposed to improve MIM by using an auxiliary trainable BEiT to degrade the images. In contrast, our paper not only alters the degradation technique but also presents a novel framework that leverages the pretrained model to defend adversarial attacks beyond merely using pretrained representations.
> > >
> > > In Section 2.1 (lines 78-85), we've already distinguished our work from other research employing different degradation methods and talked about their inspiration for our exploration into NIM. In our final paper, we will cite CIM and include the discussion here for a more comprehensive related works section.
> > >
> > > > Moreover, different research purpose does not distinguish the two works. For instance, applying the idea of MAE to audio pre-training is an application but not an innovation.
> > > >
> > >
> > > First, we would like to argue that different research purposes usually do distinguish two works. Taking the example of Vision Transformer [1], the authors claimed that they “have explored the direct application of Transformers to image recognition” (see Conclusions in [1]). Yet the paper has become one of the most important works in machine vision. Another good example is “a conceptually simple extension of Masked Autoencoders (MAE) to spatiotemporal representation learning from videos” [2], which was accepted by this venue last year and became influential in its domain.
> > >
> > > Moreover, our paper is clearly not a simple application of CIM (or any other MIM variants) to the field of adversarial robustness. Regarding the example proposed by the reviewer, it is imaginable that a completely direct application of MAE to audio could result in unexciting work, but in the case of our work, the vast disparity between the domains of adversarial robustness and image recognition makes it unlikely to directly apply CIM or its concepts to adversarial robustness.
> > >
> > > In light of these factors, we insist that our work is very different from CIM.
> > >
> > > > The pre-training objective of CIM is to denoise the corrupted input image, which is exactly the same idea proposed by the authors in this work. However, although the architectural designs and corruption approaches might be different, the underlying idea is the same.
> > > >
> > >
> > > A closer look at CIM reveals that its objective diverges from ours. CIM studied two pre-training objectives, one of which is to recover the images corrupted by a small trainable BEiT generator where a pre-trained frozen image tokenizer encoder and decoder are involved, and the other is a discriminative one. Notably, terms like "denoise" or "noise" aren't explicitly mentioned within the CIM paper. In sum, the objective of CIM is much different than “denoise the corrupted input image”.
> > >
> > > It's worth noting that even if NIM shares some similarities with CIM, equating the two as having the "same idea" is an oversimplification. If one were to perceive CIM and NIM as identical in conception, then by extension, many MIM variants, like MFM [4] and even CIM, would merely be replicas of the original MIM concept and, therefore, lack novelty. Even MIM can be seen as not novel because the idea is the “same” as MLM. However, this isn't the prevailing perspective in our scientific community, and rightfully so. As Professor Michael Black aptly articulated [3], “Taking an existing network and replacing one thing is better science than concocting a whole new network just to make it look more complex.”
> > >
> > > Furthermore, it's imperative to highlight that our work isn't solely confined to NIM. We further propose $De^3$, a novel framework that effectively leverages the pretraining NIM models for providing adversarial defense to downstream models. This is a significant contribution of our work which is recognized by other reviewers as a novel and interesting method.
> > >
> > >
> > > ---
> > >
> > > [1] Dosovitskiy et al. An Image is Worth 16x16 Words: Transformers for Image Recognition at Scale. ICLR 2021.
> > >
> > > [2] Feichtenhofer et al. Masked Autoencoders As Spatiotemporal Learners. NeurIPS 2022.
> > >
> > > [3] Black. Novelty in Science: A guide for reviewers. https://perceiving-systems.blog/en/post/novelty-in-science.

---

> > > ### Author Response · Authors · 2023-08-16
> > > **Further discussion with Reviewer qarJ (Part 2/2)**
> > >
> > > > Adding Gaussian noise to the input image and asking the model to denoise during pre-training improves the model's robustness towards noise in general.
> > > >
> > >
> > > We'd like to reference Michael’s blog [3] again where he talks about the relationship between novelty and surprise: ”The novelty, however, must be evaluated *before* the idea existed… If it is easy to explain and obvious in hindsight, this in no way diminishes the creativity (and novelty) of the idea.” To the best of our knowledge, the idea of using the ‘adding-noise-then-denoising’ pretraining to enhance the model’s robustness against adversarial noise has not been proposed before. Therefore, we argue that the idea of our work is novel even it may seem obvious in hindsight.
> > >
> > > > More results on datasets with various scales and types are required to draw solid conclusions. For instance, IN-21k, iNaturalist, COCO etc.
> > > >
> > >
> > > Following your suggestion, we have shown in the rebuttal that our method is effective on CIFAR-10 besides ImageNet. We humbly disagree that it is necessary to experiment on more datasets to prove the solidity of a method. Previously published studies often substantiate their claims based on results from a handful of datasets. For example, the recent work [5] we compared in Section 5.3, shows the empirical results on CIFAR-10 and Imagenette (a subset of 10 classes from ImageNet-1K). Another related work [6] conducts experiments solely on ImageNet. Similarly, an earlier work on adversarial robustness [7] also only used ImageNet to show the empirical results. Given these precedents, we believe that our choice of ImageNet and CIFAR-10 - both of which are well-regarded and commonly used datasets in the community - provides a sufficient foundation to support our conclusions.
> > >
> > > ---
> > >
> > > [3] Black. Novelty in Science: A guide for reviewers. https://perceiving-systems.blog/en/post/novelty-in-science.
> > >
> > > [4] Xie et al. Masked Frequency Modeling for Self-Supervised Visual Pre-Training. ICLR 2023.
> > >
> > > [5] Mo et al. When adversarial training meets vision transformers: Recipes from training to architecture. NeurIPS 2022.
> > >
> > > [6] Kong and Zhang. Understanding Masked Image Modeling via Learning Occlusion Invariant Feature. CVPR 2023.
> > >
> > > [7] Xie et al. Feature Denoising for Improving Adversarial Robustness. CVPR 2019.

---

> > > > ### Comment · Reviewer_qarJ · 2023-08-18
> > > >
> > > > * Similarity. The similarity between CIM and your proposed method does not magically disappear because you claim or emphasize that there is no similarity. For any self-supervised pre-training, the pre-text task is a crucial factor. Since both CIM and the proposed NIM denoise, the authors should at least compare and state the similarities and differences in their paper. However, they failed to do so.
> > > >
> > > > * Comparison. The authors compared with MIM and simMIM. Yet the authors claimed that MIM, simMIM and CIM all seek to learn s to learn better pretrained visual representations while NIM generative pretraining paradigm can provide adversarial robustness. Why compare with and MIM and simMIM in the first place but not some other pre-training methods designed for adversarial robustness?
> > > >
> > > > * Novelty. I mentioned CIM only because the authors compare with MIM and simMIM. Besides, NIM also share with same idea with denoising autoencoder [1]. The authors are suggested to compare with [1] using same encoder as pre-training.
> > > >
> > > > * More experiments. The authors claimed to propose a pretraining paradigm. A generic paradigm should work with various backbones and datasets. The authors compared with pre-training methods like MIM and simMIM. Then they should at least have the same scale of experiments. Meaning they should at least use the same number of backbones as MIM and simMIM. [6] mentioned by the authors are explainability work but not NIM. It is acceptable for [6] to have less experiments since its contribution is explainability. However, NIM requires more experiments to show effectiveness. [7] is a paper not in the self-pre-training domain. The standards from different domains are different. Also, [7] is an outdated paper.  In recent years, more solid experiments are required by the community.
> > > >
> > > > * BTW, it is very improper for the authors to quote a blog lecturing the reviewer on how to be a reviewer. I feel very offended by such action. Please keep the discussion professional.
> > > >
> > > >
> > > > Referecenes:
> > > >
> > > > [1] Vincent, Pascal, et al. "Extracting and composing robust features with denoising autoencoders." Proceedings of the 25th international conference on Machine learning. 2008.

---

> > > > > ### Author Response · Authors · 2023-08-21
> > > > > **Further discussion with Reviewer qarJ (Part 1/2)**
> > > > >
> > > > > > Similarity. The similarity between CIM and your proposed method does not magically disappear because you claim or emphasize that there is no similarity. For any self-supervised pre-training, the pre-text task is a crucial factor. Since both CIM and the proposed NIM denoise, the authors should at least compare and state the similarities and differences in their paper. However, they failed to do so.
> > > > > >
> > > > >
> > > > > We appreciate your suggestion and would like to state the similarities and differences between NIM and CIM here and will include the comparison in our final paper.
> > > > >
> > > > > Similarities:
> > > > >
> > > > > - Conceptually, both CIM and NIM are MIM variants, i.e., generative pretraining paradigms that replace the masking operation in MIM with some other degradation method.
> > > > >
> > > > > Differences:
> > > > >
> > > > > - The degradation methods in CIM and NIM are different.
> > > > >     - In CIM, the input images are corrupted by a generator, which consists of a pretrained frozen image tokenizer consisting of a paired encoder and decoder, and a small trainable BEiT. In particular, tokens at masked positions that are sampled according to the small BEiT output distribution together with the golden tokens that are directly produced by the image tokenizer encoder at non-masked positions constitute the input for the image tokenizer decoder. And the decoder then maps the visual tokens to corrupted images.
> > > > >     - In NIM, the input images are corrupted by Gaussian noises.
> > > > > - The pre-text tasks of CIM and NIM are different.
> > > > >     - In CIM, the authors study two pre-text tasks, ResPix and RevDet. ResPix is a generative task where the enhancer predicts the uncorrupted pixel value for all positions given the corrupted images, and RevDet is a discriminative task where the enhancer determines whether each visual token is replaced by a generator sample or not.
> > > > >     - In NIM, the pre-text task is to denoise the added Gaussian noise.
> > > > > - The hyperparameter that controls the difficulty of the pre-text task is different.
> > > > >     - In CIM, the difficulty of the pre-text task mainly depends on the masking ratio, which is set to 50% globally in a pre-determined way.
> > > > >     - In NIM, the difficulty of the pre-text task depends on the $\sigma$ of the added Gaussian noise, which is sampled from a $\Gamma$ distribution.
> > > > > - CIM and NIM are used in different ways in the two papers.
> > > > >     - CIM is only used for providing pretrained visual representations. Only the enhancer is used for downstream fine-tuning and the generator is thrown away.
> > > > >     - NIM is used not only for providing pretrained features but also for adversarial defense. Under our proposed novel framework, $De^3$, both encoder and decoder are utilized for enhancing the adversarial robustness of downstream models.
> > > > >
> > > > > > Comparison. The authors compared with MIM and simMIM. Yet the authors claimed that MIM, simMIM and CIM all seek to learn s to learn better pretrained visual representations while NIM generative pretraining paradigm can provide adversarial robustness. Why compare with and MIM and simMIM in the first place but not some other pre-training methods designed for adversarial robustness?
> > > > > >
> > > > >
> > > > > We compare with representative MIM pretraing methods because the goal of this work is to show that NIM can achieve a stronger and tunable accuracy-robustness trade-off compared to MIM. We have also compared our NIM+$De^3 method with a recent adversarial training method in Section 5.3 to help understand the effectiveness of our method.
> > > > >
> > > > > > Novelty. I mentioned CIM only because the authors compare with MIM and simMIM. Besides, NIM also share with same idea with denoising autoencoder [1]. The authors are suggested to compare with [1] using same encoder as pre-training.
> > > > > >
> > > > >
> > > > > While we hope to show the experimental results comparison with CIM as we did with MAE and SimMIM, we cannot find any code implementing CIM, official or unofficial, and it is very challenging to reimplement CIM within this short time of the discussion period.
> > > > >
> > > > > As we have discussed in lines 127-130, the term “denoising” in DAE [1] means recovering masked pixels, instead of removing added Gaussian noise as in NIM. Following your suggestion, here we show a performance comparison with DAE using the same encoder. Since the time is limited, we conduct the experiments on CIFAR-10, following again the implementation of [a], and only pretrained DAE and NIM-MAE for 200 epochs with 20 epochs for warmup. The results are as follows:
> > > > >
> > > > > | Model | DAE | DAE | NIM-MAE | NIM-MAE |
> > > > > | --- | --- | --- | --- | --- |
> > > > > | $De^3$ | None | $\gamma$=0.75 | $\sigma$=40 | $\sigma$=70 |
> > > > > | Clean | 71.44 | 52.89 | 76.44 | 71.20 |
> > > > > | FGSM | 5.14 | 26.03 | 25.98 | 35.78 |
> > > > > | PGD-10 | 0.05 | 8.08 | 15.15 | 26.88 |
> > > > > | AA | 0.00 | 5.06 | 16.17 | 27.62 |
> > > > >
> > > > > It is shown that NIM outperforms DAE in both clean accuracy and adversarial robustness.
> > > > >
> > > > > ---
> > > > > [a] https://github.com/IcarusWizard/MAE.

---

> > > > > ### Author Response · Authors · 2023-08-21
> > > > > **Further discussion with Reviewer qarJ (Part 2/2)**
> > > > >
> > > > > > More experiments. The authors claimed to propose a pretraining paradigm. A generic paradigm should work with various backbones and datasets. The authors compared with pre-training methods like MIM and simMIM. Then they should at least have the same scale of experiments. Meaning they should at least use the same number of backbones as MIM and simMIM. [6] mentioned by the authors are explainability work but not NIM. It is acceptable for [6] to have less experiments since its contribution is explainability. However, NIM requires more experiments to show effectiveness. [7] is a paper not in the self-pre-training domain. The standards from different domains are different. Also, [7] is an outdated paper. In recent years, more solid experiments are required by the community.
> > > > > >
> > > > >
> > > > > In MAE, the authors conduct experiments with ViT-B, ViT-L, and ViT-H as backbones, and we have also shown that our method is effective on these backbones. Besides, we have followed your suggestion and shown that our method is effective on CIFAR-10 besides ImageNet. While [6] performed some theoretical analysis, its main contributions are based on experimental results, so sufficient experiments are needed to prove their findings solid. Also, we hope the reviewer may consider the explanation in our rebuttal that performing adversarial attacks for other tasks than classification is beyond this paper’s scope. Overall, we would like to clarify again that the purpose and novelty of our paper are not to propose a new pretraining paradigm, but to show the superiority of NIM over MIM in terms of adversarial robustness using the proposed defense framework, and we believe that our experimental results are sufficient to prove this point.
> > > > >
> > > > > > BTW, it is very improper for the authors to quote a blog lecturing the reviewer on how to be a reviewer. I feel very offended by such action. Please keep the discussion professional.
> > > > > >
> > > > >
> > > > > We deeply apologize for the unintended offense caused by referencing the blog. It was never our intention to lecture or undermine your expertise. Our intent was solely to share an understanding of novelty and believe that quotations from influential scholars may make such an understanding more convincing. We notice that discussions on OpenReview sometimes reference external articles or blogs. For example, in [b, c], the same blog was referenced by the authors. Nevertheless, we recognize the importance of ensuring our discussion remains respectful and professional at all times. Once again, we sincerely regret the oversight and appreciate your feedback.
> > > > >
> > > > > ---
> > > > > [b] https://openreview.net/forum?id=2OdAggzzF3z&noteId=0X9_fcsILKT
> > > > >
> > > > > [c] https://openreview.net/forum?id=nxl-IjnDCRo&noteId=JqTRaHS43ww

---

> ### Author Response · Authors · 2023-08-14
> **Further Discussion with Reviewer qarJ**
>
> Dear Reviewer qarJ,
>
> We genuinely appreciate the time and effort you've invested in reviewing our paper. We have carefully provided relevant responses and results to your concerns. We are eager to further discuss with you and gain your insights. Please let us know if any aspect of our work remains unclear or if you have additional feedback. Thank you.
>
> Warm regards,
>
> Authors

---

> ### Comment · Area_Chair_RrHr · 2023-08-17
> **Please take a look at authors' responses and other reviewers' comments**
>
> Dear Reviewer,
>
>  Please take a look at authors' responses and other reviewers' comments, Thank you very much.
>
> BTW, for a solid review, it would be better to give more details to make a decesion. The current review seems to be short.

---

### Official Review · Reviewer_XF5y · 2023-07-03

**Soundness:** 3 good
**Presentation:** 4 excellent
**Contribution:** 3 good
**Rating:** 7
**Confidence:** 3

**Summary:**

The authors introduce noise image modeling (NIM) as a self-supervised pretext task and demonstrate that their encoder-decoder architecture decreases the success of adversarial attacks by adding noise to a perturbed image and then denoising it. Their method is called $De^3$. Furthermore, the author conduct a major comparison effort to masked image modeling (MIM) and the capabilities against adversarial attacks.


**Strengths:**

- This paper is well written and easy to understand.
- "flooding out" adversarial attacks is an interesting idea.
- The method utilizes the usually unused decoder for "cleaning" adversarial images.
- $De^3$, as an adversarial prevention method, can be applied without generating much computation overhead during inference or training.
- NIM with $De^3$ can use a dynamic trade-off between clean accuracy and the vulnerability to adversarial images.
- On-par results with MIM while doing adversarial training.

**Weaknesses:**

1. Table 1/Figure 3:
- The "flooding out effect" for very large sigma (140) seems not reasonable, since it corrupts the whole image. How strong is the flooding in comparison to the adversarial perturbation? I think it is necessary to show the scale difference of (i) the adversarial attacks and (ii) the added noise.
- I am a little bit surprised by the denoising performance with high noise-levels (see Figure 3 (f) - sigma 140):  the reconstruction seems "too good to be true", where it even recovers the smallest features. This seems odd despite other efforts [1]. I think it needs further extensive investigation:  (i) when does the model break in terms of reconstruction quality? What happens when you use an even larger value for sigma? (ii) when does the adversarial defense fail when a high-noise level (sigma 100, 150, 200) is used?
2. Figure 2 (a) needs captions for the image rows/columns to understand it better.

3. Table 1: Inconsistent use of bold highlighting?

4. Table 1: I am not sure if this is a fair comparison, since NIM utilizes its built-in adversarial prevention
           while MAE/SimMIM (MIM) has none (gamma is not implicitly designed for dealing with adversarial images).
5. line 264-268: From the readers perspective, it seems that the authors tried to achieve state-of-the-art with $De^3$ but failed, so they redirect their writing to fit a different goal. I suggest, the authors omit the sentences about not wanting to achieve SOTA.
6. Figure 5: I suggest to use more symbols for different sigmas. Differentiate between fixed/random sigma and MAE baseline with triangle and then use square/diamond/circle.
7. Figure 5: Ordering of the entries in the legend can be improved

References:

[1] Mahdaoui, Assia El, Abdeldjalil Ouahabi, and Mohamed Said Moulay.
"Image denoising using a compressive sensing approach based on regularization constraints."
Sensors 22.6 (2022): 2199.

**Questions:**

Do you think that there is a way to construct strong adversarial examples if the attacker knows that the NIM architecture is used, despite the noise randomness?


**Limitations:**

Limitations are not discussed in the paper.

---

> ### Author Rebuttal · Authors · 2023-08-09
>
> We thank the reviewer for the constructive comments. Here are our responses:
>
> > The "flooding out effect" for very large sigma (140) seems not reasonable, since it corrupts the whole image.
>
> > The reconstruction seems "too good to be true". This seems odd despite other efforts [1].
>
> Given that noisy images with large sigma become completely meaningless to humans, it is understandable that one may think the reconstruction is “too good to be true”. However, the experimental results show that ViTs are indeed capable of removing intense Gaussian noise. Note that we are denoising artificial Gaussian noise, which is much easier than real-world unknown noise in [1]. Table 2 (c) in [2] shows a similar result: Top-1 acc for $\sigma=100$ is only lower for 0.1% than the best $\sigma=75$. We think the main reason that highly noisy images are meaningless to humans is that human eyes have limited capabilities, rather than that the information is completely corrupted.
>
> > How strong is the flooding in comparison to the adversarial perturbation? I think it is necessary to show the scale difference.
>
> In Table 1, when the images are not normalized (the range is [0, 255]), the Gaussian noise’s $\sigma$ is 70 (or 140), and the budget of adversarial perturbations is 4. In other words, the magnitude of the Gaussian noise we added is about 70/4=17.5 (or 35) times the adversarial noise. Thank you for this valuable advice and we will add this scale difference to our final paper.
>
> > When does the model break in terms of reconstruction quality? What happens when you use an even larger value for sigma? When does the adversarial defense fail when a high-noise level (sigma 100, 150, 200) is used?
>
> When using larger values for sigma, the clean accuracy will decrease, and the robust accuracy will also decrease when it gets too large. For example, in Figure 4, accuracy against FGSM gets lower when the $\sigma$ is larger than 100. When $\sigma$ gets too high (e.g., 200 or 250), the reconstruction quality will be bad, leading to low clean and robust accuracy. Here are the results when using very large $\sigma$ in $De^3$:
>
> | $De^3$ | $\sigma=150$ | $\sigma=200$ | $\sigma=250$ |
> | --- | --- | --- | --- |
> | Clean | 69.09 | 61.58 | 36.84 |
> | FGSM | 52.19 | 47.24 | 28.28 |
> | PGD-10 | 39.61 | 37.37 | 23.63 |
>
> > Figure 2 (a) needs captions for the image rows/columns to understand it better.
>
> Thank you for the advice. In the top row, the four images are noisy images of $\sigma$=0 (original image), 50, 75, and 100, respectively. In the middle, the four are reconstructed images from the corresponding noisy images at the top row by the NIM model whose $\sigma$ in pretraining is globally set to 75, and the bottom row is four reconstructed images from the corresponding noisy images at the top row by the NIM model whose $\sigma$ in pretraining is randomly sampled from $\Gamma(25, 3)$. We will make it clearer in the final version.
>
> > Table 1: Inconsistent use of bold highlighting?
>
> We highlight the highest performance for each setting among variants of the same MIM models. In other words, **83.05** is highlighted because it is the highest clean accuracy achieved by all *MAE* or *NIM-MAE* methods, and **51.84** is highlighted because it is the highest accuracy against FGSM achieved by all *SimMIM* or *NIM-SimMIM* methods.
>
> > Table 1: I am not sure if this is a fair comparison, since NIM utilizes its built-in adversarial prevention while MAE/SimMIM (MIM) has none.
>
> We’d like to clarify that NIM doesn’t have any “built-in adversarial prevention”, either, because it is trained to denoise *Gaussian noise*, instead of specific *adversarial noise*. The comparison is fair because, from pretraining and fine-tuning to evaluation, we ensure everything is the same and the only variable is the degradation method.
>
> > line 264-268: I suggest, the authors omit the sentences about not wanting to achieve SOTA.
>
> Our intention to write these sentences was to prevent misleading the readers to get the idea from the comparison with adversarial training methods that our purpose was to chase the SOTA. We included the comparison to provide readers with a comprehensive understanding of our method's effectiveness. As stated in the Introduction, our primary goal is to explore how the generative pretraining paradigm can provide adversarial robustness beyond pretrained visual features, and our main takeaway for the community is that NIM can achieve a strong and tunable accuracy-robustness trade-off that MIM models are unable to, indicating the superiority of NIM over MIM in terms of adversarial robustness. However, taking your feedback into account, we'll consider omitting or rephrasing the sentences to prevent any potential misunderstandings.
>
> > Figure 5: I suggest to use more symbols for different sigmas.
>
> > Figure 5: Ordering of the entries in the legend can be improved.
>
> We genuinely appreciate your constructive feedback on the representation in Figure 5. We will incorporate these modifications in our final paper.
>
> > Do you think that there is a way to construct strong adversarial examples if the attacker knows that the NIM architecture is used, despite the noise randomness?
>
> First, we’d like to clarify that our experiments are under this very assumption that the attacker has full access to both the classifier and the defense model, only without knowing the sampling result of the random noise.
>
> To answer the question, we do believe there may be ways to break the defense provided by NIM and $De^3$. For example, as stated in the Limitation section, we only prove that our method is effective against noise-based attacks, but other forms of attacks, like adversarial patches, might present challenges to our defense mechanism.
>
> > Limitations are not discussed in the paper.
>
> Please refer to the Section B of our supplementary material.
>
> ---
>
> [2] Xie et al. Masked Frequency Modeling for Self-Supervised Visual Pre-Training. 2022.

---

> ### Author Response · Authors · 2023-08-14
> **Further Discussion with Reviewer XF5y**
>
> Dear Reviewer XF5y,
>
> We genuinely appreciate the time and effort you've invested in reviewing our paper. We have carefully provided relevant responses and results to your concerns. We are eager to further discuss with you and gain your insights. Please let us know if any aspect of our work remains unclear or if you have additional feedback. Thank you.
>
> Warm regards,
>
> Authors

---

> > ### Comment · Reviewer_XF5y · 2023-08-15
> >
> > Thanks for your responses and the clarifications. My concerns were addressed in an appropriate way. Overall, NIM with De^3 seem to be interesting, novel and competitive with existing related methods. My main issue was the reconstruction quality for high sigmas which apparently seems to work. Hence, I am increasing my rating to 7.

---

### Official Review · Reviewer_ovhG · 2023-07-04

**Soundness:** 2 fair
**Presentation:** 3 good
**Contribution:** 3 good
**Rating:** 5
**Confidence:** 3

**Summary:**

The paper presents a straightforward approach for defending against adversarial attacks while incorporating pre-trained feature learning through the utilization of noisy images. Inspired by Masked Image Modeling (MIM), the paper replaces the masking pretext task by introducing a substantial amount of noise into the image. Subsequently, a transformer-based encoder/decoder is trained to reconstruct the original image, and the encoder is fine-tuned to acquire features that are invariant to noise by comparing the encoded features of the original image input with those of the reconstructed images. The defense mechanism involves retaining the decoder for the downstream task, employing an [encoder → decoder → encoder] mechanism to obtain features that can be further fine-tuned for the downstream task.

The authors perform experiments on ImageNet-1K using the ViT-Base backbone and evaluate the proposed method against several existing adversarial attacks, including $l_{\infty}$ bound attacks, FGSM, PGD, and AutoAttack (AA). The results demonstrate that the proposed method exhibits greater robustness compared to baseline pre-trained methods like MAE or SIMM.


**Strengths:**

The finding of the paper is interesting, which suggests that the adversarial attacks present as a kind of noise, and it makes sense that denoising works to some extent. This aspect might open up possibilities for combining existing works to further enhance downstream accuracy and adversarial defense.

The paper writing is easy to follow and the proposed method is simple and comprehensive.


**Weaknesses:**

1. It would be beneficial to have a more extensive comparison by including well-known self-supervised features like SimCLR, MOCO, DINO, and others, to assess the robustness of the features learned by these methods in comparison to the proposed method.

2. To enhance the paper, it would be valuable if the authors conducted experiments on another dataset and included more baseline methods to gain insights into the generalizability of the proposed method.

3. It would be valuable to compare the proposed method with existing adversarial defense approaches, such as GAN-based methods.

4. Despite some improvements, the defense results are not particularly impressive and do not appear to completely eliminate the attacks. This suggests that the attacks might involve more than just noise.

5. It should be noted that using additional decoders could be a drawback depending on the applications, as it significantly increases memory usage during inference.

The approach seems novel, and I find the current version of the paper is fine unless there are serious concerns / flaws from other reviewer’s feedback. However, I would recommend conducting additional experiments to further showcase the generalizability of the proposed method.


**Questions:**

I wonder if the authors have considered keeping the original MAE approach and combining it with the noisy modelling technique to explore any potential further improvements?

See questions above also.

---

> ### Author Rebuttal · Authors · 2023-08-09
>
> Thank you for your insightful comments and suggestions! Our answers are as follows:
>
> > It would be beneficial to have a more extensive comparison by including well-known self-supervised features like SimCLR, MOCO, DINO, and others.
>
> While we concur that the comparison would be a great direction to explore, we would like to highlight that it is beyond the scope of this paper. The primary goal of this work is to explore how the generative pretraining paradigm can provide adversarial robustness beyond visual features. Therefore, other pretraining frameworks like contrastive learning (CL) are not considered as they are not applicable to our proposed $De^3$ defense.
>
> However, we believe this could be a great research direction for future work. Existing works have compared ViTs and CNNs' adversarial robustness [1], and also compared the pretrained visual features of MIM and CL [2], but the comparison between MIM and CL from the perspective of adversarial robustness has not been explored. This research gap could be a good starting point for a valuable research project.
>
> > To enhance the paper, it would be valuable if the authors conducted experiments on another dataset and included more baseline methods.
>
> Thanks for this valuable suggestion. We conducted another experiment on CIFAR-10, based on the implementation of [3]. We set the attack radius $\epsilon = 8/255$, and other settings remain the same as in Table 1. The results are as follows:
>
> | Model | MAE | MAE | NIM-MAE | NIM-MAE | NIM-MAE |
> | --- | --- | --- | --- | --- | --- |
> | $De^3$ | None | $\gamma$=0.75 | None | $\sigma$=40 | $\sigma$=70 |
> | Clean | **89.88** | 78.36 | 88.31 | 81.09 | 76.30 |
> | FGSM | 17.51 | **65.13** | 21.66 | 43.26 | 48.86 |
> | PGD-10 | 0.01 | 22.50 | 2.77 | 32.24 | **40.26** |
> | AA | 0.00 | 6.220 | 0.00 | 31.49 | **41.20** |
>
> Interestingly, we observe that unlike on ImageNet, MAE provides a stronger defense against FGSM on CIFAR-10. The reasons are probably that 1) for low-resolution images, MAE can generate good reconstructions from masked images; 2) when using MAE in $De^3$, 75% of adversarial perturbations will be masked, and if the attack is weak like FGSM, the rest of the perturbation would be too weak to break the model. However, for stronger attacks, NIM still provides better defense.
>
> Regarding the baseline method, we adopted two representative MIM methods, MAE and SimMIM, as two baseline methods. It is observed that NIM provides consistently better adversarial defense than MIM, shown in Table 1 of our main paper.
>
> > It would be valuable to compare the proposed method with existing adversarial defense approaches.
>
> In Section 5.3, we show the comparison of our method and a recent adversarial training approach where ViTs are also adopted as the backbone. Until recently, most adversarial defense methods are based on CNNs and the comparison with them would be out of this work’s scope. More importantly, we would like to emphasize that the main point of this work is not to compete with other adversarial defense methods, but to show that NIM can achieve a strong and tunable accuracy-robustness trade-off and to demonstrate to the community that NIM can serve as a promising and advantageous self-supervised learning paradigm.
>
> > Despite some improvements, the defense results are not particularly impressive and do not appear to completely eliminate the attacks.
>
> So far, defending against adversarial attacks is still a very challenging task and there is no method that is completely immune to attacks. We show that our proposed method can achieve an improvement in robustness against PGD attack by 34.36%, with a marginal clean accuracy drop of 4.37%. Given that our model is trained to denoise *Gaussian noise*, instead of *adversarial noise*, we think the improvement is significant and interesting.
>
> > This suggests that the attacks might involve more than just noise.
>
> Indeed, even when $\sigma$ = 250 (62.5 times the adversarial perturbation budget), the clean accuracy (36.84%) is still higher than the robust accuracy against PGD attack (23.63%). This result shows that even when adding a very large Gaussian noise, the adversarial noise would not be fully flooded and removed, indicating that adversarial perturbation may be meaningful to the neural networks, rather than just noise.
>
> > It should be noted that using additional decoders could be a drawback depending on the applications, as it significantly increases memory usage during inference.
>
> Yes, incorporating additional decoders can indeed increase memory requirements, which could be a concern, especially for applications with tight memory constraints. As suggested by Reviewer judd, distillation could be a potential solution. By training a smaller network to mimic the behavior of the larger network, we could obtain a more efficient model that retains much of the adversarial robustness of the original approach. We acknowledge this limitation and appreciate the insightful suggestion, and we're keen to investigate this avenue in future research.
>
> > I wonder if the authors have considered keeping the original MAE approach and combining it with the noisy modelling technique?
>
> Thank you for providing this great idea! We have not considered this direction, but we believe it has great potential. It reminds us of the findings presented in [4], where the combination of zooming-in and masking outperformed using only masking. Therefore, it is reasonable to expect that combining masking and adding noise would bring further improvements. We will certainly consider investigating this direction in subsequent studies. Thanks again for the advice.
>
> ---
> [1] Bai et al. Are Transformers More Robust Than CNNs? NeurIPS 2021.
>
> [2] Wei et al. Contrastive Learning Rivals Masked Image Modeling in Fine-tuning via Feature Distillation. 2022.
>
> [3] https://github.com/IcarusWizard/MAE.
>
> [4] Tian, et al. Beyond Masking: Demystifying Token-Based Pre-Training for Vision Transformers. 2022.

---

> ### Author Response · Authors · 2023-08-14
> **Further Discussion with Reviewer ovhG**
>
> Dear Reviewer ovhG,
>
> We genuinely appreciate the time and effort you've invested in reviewing our paper. We have carefully provided relevant responses and results to your concerns. We are eager to further discuss with you and gain your insights. Please let us know if any aspect of our work remains unclear or if you have additional feedback. Thank you.
>
> Warm regards,
>
> Authors

---

### Official Review · Reviewer_judd · 2023-07-06

**Soundness:** 4 excellent
**Presentation:** 3 good
**Contribution:** 4 excellent
**Rating:** 8
**Confidence:** 5

**Summary:**

This paper proposes a novel adversarial defense method $De^3$ to utilize the strong denoising ability of NIM models. The proposed method first adds some Gaussian noise to the adversarial samples and then tries to reconstruct the original images. Experiments show the advantage of NIM over MIM in terms of adversarial robustness.

**Strengths:**

1. The idea of NIM is interesting.  The proposed method obtains the adversarial defense as well as provides pretrained features by a simple yet effective modification on masked image modeling.
2. The idea of De3 is instructive to enhance adversarial defense through reconstructing clean images from intensely noisy images.
3. The experiments demonstrate that the proposed method can achieve comparable defense performance with adversarial training.
4. This paper is well-written and easy to follow.


**Weaknesses:**

1. Although De3 can enhance adversarial defense, it increases computational cost during inference time. Note that there exists two encoders  and one decoder in De3, can we distill another encoder  from them and use only it during inference time?
2. This paper compares NIM against MIM in terms of accuracy and adversarial robustness. It seems that MIM achieves better accuracy on clean images. I wonder can we combine them together and obtain a better trade-off between accuracy and robustness?

**Questions:**

I wonder the difference of robustness against other types of attacks, like black-box ones.

**Limitations:**

Limitations are adequately discussed in the supplemental materials.

---

> ### Author Rebuttal · Authors · 2023-08-10
>
> Thank you very much for your positive affirmation of our work and your constructive suggestions. Here are our responses.
>
> > Although De3 can enhance adversarial defense, it increases computational cost during inference time. Note that there exists two encoders and one decoder in De3, can we distill another encoder from them and use only it during inference time?
> >
>
> We appreciate this valuable suggestion. As we acknowledged in our Limitation section, the computational cost introduced by the defense process is a major drawback of our method. We have not considered using distillation for resolving the issue, but we believe it could be a direction with great potential. By training a smaller network to mimic the behavior of the original network, we could obtain a more efficient model that retains much of the adversarial robustness while lowering the computational costs. Moreover, given that one encoder essentially stems from the initialization of the other before fine-tuning, it's plausible that there's room for optimization by either merging or compressing them. We're genuinely grateful for this constructive feedback and will look into this possibility in our subsequent research.
>
> > This paper compares NIM against MIM in terms of accuracy and adversarial robustness. It seems that MIM achieves better accuracy on clean images. I wonder can we combine them together and obtain a better trade-off between accuracy and robustness?
> >
>
> Thank you for another insightful idea. Indeed, since MIM exhibits better accuracy on clean images and NIM shows stronger robustness, there could be potential in combining them to achieve a better accuracy-robustness trade-off. Previous work [1] shows that when combining zooming-in and masking, the pretraining performance would be higher than using only masking. Therefore, it is reasonable to expect that combining masking and adding noise would bring further improvements. We will certainly delve into this promising avenue in our future work.
>
> > I wonder the difference of robustness against other types of attacks, like black-box ones.
> >
>
> We adopt Square [2] as a example of black-box attacks, and set the perturbation budget $\epsilon$=16/255 and a budget of 1,000 queries. The results are as follows compared with white-box PGD-10 attacks.
>
> | Model | MAE | MAE | NIM-MAE | NIM-MAE | NIM-MAE |
> | --- | --- | --- | --- | --- | --- |
> | $De^3$ | None | $\gamma$=0.75 | None | $\sigma$=70 | $\sigma$=140 |
> | Clean | 83.05 | 44.96 | 82.58 | 78.68 | 70.69 |
> | PGD-10 (WB) | 0.25 | 11.58 | 0.31 | 34.61 | 39.82 |
> | Square (BB) | 1.17 | 33.59 | 2.44 | 73.14 | 67.29 |
>
> It is shown that our $De^3$ with NIM is very effective for defending against this black-box attack.
>
> ---
>
> [1] Tian, et al. Beyond Masking: Demystifying Token-Based Pre-Training for Vision Transformers. 2022.
>
> [2] Andriushchenko et al. Square attack: a query-efficient black-box adversarial attack via random search. ECCV 2020.

---

> > ### Comment · Reviewer_judd · 2023-08-18
> >
> > I appreciate the authors for dealing with my concerns and including more experiments. The additional results make me more convinced.

---

### Official Review · Reviewer_P9xe · 2023-07-10

**Soundness:** 4 excellent
**Presentation:** 4 excellent
**Contribution:** 4 excellent
**Rating:** 8
**Confidence:** 4

**Summary:**

This paper proposes a method called Noisy Image Modeling (NIM) as self-supervised learning to improve the adversarial robustness of pretrained features. NIM uses denoising as a pretext task and is effective in reconstructing noisy images for representation learning. In addition, the authors propose a defense technique called De^3 that leverages the denoising capability of NIM to enhance robustness against adversarial attacks. Experimental results show that NIM with De3 defense outperforms Masked Image Modeling (MIM) in terms of adversarial robustness while maintaining competitive performance on clean data. The paper concludes by highlighting the potential of NIM and other variants of MIM for generative visual pretraining.

**Strengths:**

1. Different from the popular MIM, NIM improves the pretrained features by resisting adversarial attacks. The experiment verfies the effectiveness of the proposed method.
2. In addition to the representation learning method, the paper proposes a defense technique called De3 that utilizes the denoising capability of NIM to enhance robustness against adversarial attacks.
3. The paper compares the performance of NIM with the proposed De3 defense to Masked Image Modeling (MIM) and other adversarial training methods. Experimental results show that NIM with De3 defense outperforms MIM in terms of adversarial robustness while maintaining competitive performance on clean data.
4. The paper introduces a modification to NIM that allows for a tunable trade-off between accuracy and robustness.

**Weaknesses:**

1. The paper does not thoroughly analyze the computational cost of the proposed method. Adversarial training methods, such as the one proposed by Wang et al. [34], are known to be computationally expensive. It would have been valuable to compare the computational cost of NIM with other defense techniques.
2. The paper does not extensively explore the impact of hyperparameters on the performance of the proposed method. For example, the authors mention that the trade-off between clean accuracy and robustness can be adjusted by varying the noise level hyperparameter, but they do not provide a detailed analysis of the optimal values or the sensitivity of the method to different hyperparameter settings.
3. The paper does not analyze the transferability of adversarial examples between different models. It would have been valuable to investigate whether the robustness achieved by NIM with De^3 defense can generalize to other models or if it is specific to the pretrained model used in the experiments [1].

**Questions:**

1. Computational cost.
2. Parameter Sensitivity Analysis.
3. The generalization of the proposed  method.

**Limitations:**

Yes

---

> ### Author Rebuttal · Authors · 2023-08-10
>
> We appreciate your approval and recognition of this paper. Our answers are listed below.
>
> > The paper does not thoroughly analyze the computational cost of the proposed method. Adversarial training methods, such as the one proposed by Wang et al. [34], are known to be computationally expensive. It would have been valuable to compare the computational cost of NIM with other defense techniques.
>
> Thank you for this valuable suggestion. Indeed, our paper could have benefitted from a comparison of the computational costs between the proposed NIM approach and existing adversarial training methods. We evaluated the throughput of the adversarial training method [21] (used in the comparisons of Section 5.3) and our proposed NIM approach, using an identical training framework implemented on 8 A100 GPUs. The result is as follows
>
> | Method | Throughput (images/sec) | Clean | PGD-10 |
> | :---: | :---: | :---: | :---: |
> | AT-ViT [21] | 488.0 | 69.28 | 39.97 |
> | Ours | 2009.4 | 70.69 | 39.82 |
>
> Note that the adversarial training method is trained for 20 epochs and ours are trained for 100 epochs. Therefore, adjusted for epochs, our method are (488.0/20)/(2009.4/100)=1.2 times faster than the adversarial training method.
>
> > The paper does not extensively explore the impact of hyperparameters on the performance of the proposed method. For example, the authors mention that the trade-off between clean accuracy and robustness can be adjusted by varying the noise level hyperparameter, but they do not provide a detailed analysis of the optimal values or the sensitivity of the method to different hyperparameter settings.
> >
>
> Sorry for the confusion. Regarding the trade-off that can be adjusted by the noise level hyperparameter $\sigma$, we would like to refer to Figure 4 and the description in lines 247-261, where the variance in clean and robust accuracies with respect to $\sigma$ is explicitly presented. It is shown that by increasing the $\sigma$ in defense, the clean accuracy would decrease as the reconstruction gets poorer, but the robust accuracy would be enhanced because adversarial perturbations are flooded by stronger noises.
>
> In addition, we present how different random distributions of $\sigma$ in pretraining influence the fine-tuned models’ performance in Section 5.4 and we show that $\sigma \sim \Gamma(25,3)$ achieves the best performance among all models. It is noteworthy that due to the limitation of computational resources, it was challenging to conduct a comprehensive grid search for hyperparameter optimization. However, we would like to highlight that the main point of this work is to show that NIM can achieve a stronger and tunable accuracy-robustness trade-off compared to MIM, instead of achieving the highest possible performance with optimal hyperparameters.
>
> > The paper does not analyze the transferability of adversarial examples between different models. It would have been valuable to investigate whether the robustness achieved by NIM with De^3 defense can generalize to other models or if it is specific to the pretrained model used in the experiments [1].
> >
>
> Thanks for this insightful feedback. Indeed, exploring the transferability of the proposed adversarial defense method would be very helpful for a more comprehensive understanding of our method and its applicability. Given that our method does not train the model to be robust to some specific adversarial noise but removes the perturbations along with strong Gaussian noise, we are optimistic about the generalizability of our defense. Due to limited time and computational resources, we haven't provided empirical evidence to confirm this claim, but we will work on it once we are able to.
>
> ---
>
> [21] Mo, et al. When adversarial training meets vision transformers: Recipes from training to architecture. NeurIPS 2022.

---

### Decision · Program_Chairs · 2023-09-21

**Decision:**

Accept (poster)

**Comment:**

After the rebuttal period, most reviewers lean towards acceptance. Although reviewer qarJ raised concerns about the novelty of the method, AC aligns with most reviewers’ comments that the paper proposed an effective defense method against adversarial attack, and it allows for a tunable trade-off between accuracy and robustness. Overall this work can be of large interest to the community working on adversarial defense.